# Lipid metabolism adaptations are reduced in human compared to murine Schwann cells following injury

Sofia Meyer zu Reckendorf [1]✉, Christine Brand[2], Maria T. Pedro[3], Jutta Hegler[1], Corinna S. Schilling[1], Raissa Lerner[4], Laura Bindila[4], Gregor Antoniadis[3] & Bernd Knöll [1]✉

Mammals differ in their regeneration potential after traumatic injury, which might be caused by species-specific regeneration programs. Here, we compared murine and human Schwann cell (SC) response to injury and developed an ex vivo injury model employing surgery-derived human sural nerves. Transcriptomic and lipid metabolism analysis of murine SCs following injury of sural nerves revealed down-regulation of lipogenic genes and regulator of lipid metabolism, including *Pparg* (peroxisome proliferator-activated receptor gamma) and S1P (sphingosine-1-phosphate). Human SCs failed to induce similar adaptations following ex vivo nerve injury. Pharmacological PPARg and S1P stimulation in mice resulted in up-regulation of lipid gene expression, suggesting a role in SCs switching towards a myelinating state. Altogether, our results suggest that murine SC switching towards a repair state is accompanied by transcriptome and lipidome adaptations, which are reduced in humans.

[1] Institute of Physiological Chemistry, Ulm University, 89081 Ulm, Germany. [2] Department of Neurosurgery, Hospital Bogenhausen, 81925 Munich, Germany. [3] Peripheral Nerve Surgery Unit, Department of Neurosurgery, Ulm University, District Hospital, 89312 Günzburg, Germany. [4] Institute of Physiological Chemistry, University Medical Centre of the Johannes Gutenberg University Mainz, 55128 Mainz, Germany. ✉email: sofia.meyer-zu-reckendorf@uni-ulm.de; bernd.knoell@uni-ulm.de

Peripheral nerve injuries (PNIs) are diagnosed in 2–3% of patients admitted to trauma centres[1,2]. Although peripheral nerves have an intrinsic regeneration potential, this varies according to patients' age, injury location and severity[2]. In fact, spontaneous regeneration is limited in human nerves and even after surgical intervention, functional recovery is often poor[3]. Hence, PNIs have a strong impact on patients due to motor and sensory function loss, dysesthesias, paralysis and neuropathic pain[3].

As shown in rodents, the regenerative capacity of peripheral nerves depends on Schwann cell (SC) properties. After injury, SCs typically switch from a differentiated myelinating state into a pro-regenerative repair phenotype[4]. This SC reprograming involves changes in transcription factor (TF) expression including *cJun* up-regulation and *Egr2* (early growth response 2) down-regulation[4,5]. During reprograming, differentiated SCs stop myelin production (e.g. myelin genes like *Mbp*, myelin basic protein; *Pmp22*, peripheral myelin protein 22). Furthermore, SCs secrete cytokines and chemokines (e.g. *Ccl2*) resulting in immune system activation and produce signalling molecules and growth factors (e.g. *Shh*, sonic hedgehog; *Gdnf*, glial cell line-derived neurotrophic factor) to promote axonal outgrowth[5–7]. During this phase, SCs proliferate and form so-called Büngner bands serving as guiding tracks for outgrowing axons[5]. This acute phase following PNI, which involves SC reprograming, axon degeneration and myelin debris clearance is called Wallerian degeneration[4]. Subsequently, in a later post-injury phase, repair SCs re-differentiate into myelinating SCs as axons regenerate and re-innervate target tissue[4].

General features of SC responses in PNI are likely conserved in humans and rodents[4,8–10]. However, differences also emerge since re-growth and functional recovery is more accelerated in rodents compared to PNI patients. This is reflected by increased nerve regeneration rates in rodents (~4.5 mm/day) compared to humans (1–1.5 mm/day)[11–14]. Currently, molecular mechanisms accounting for such differential regeneration potential between species are barely identified. This is mainly due to the near absence of data on human SC responses in vivo with most data available being generated in SC cultures[15–18].

In this study, we provide a comprehensive comparison of human vs. mouse nerve tissue after PNI. For this, we establish an experimental system allowing for direct comparison of human and murine acute SC injury responses as closely as possible to in vivo conditions. Freshly dissected human and mouse sural nerves are cultured and analysed at different post-injury time points. This ex vivo injury model largely preserves 3D architecture and cell–cell interactions present in vivo. We observe that human SC injury responses are decreased compared to mice. In addition, we identify regulation of lipid metabolism as mechanism involved in SC reprograming in mice, which is delayed in human nerves. Our results emphasise the importance of identifying molecular differences between mice and humans providing novel therapeutic targets for nerve regeneration in patients.

## Results

**Characterisation of an ex vivo nerve injury model.** We analysed acute SC reactions in human and murine nerves to identify similarities and differences possibly explaining the limited regeneration capacity of human nerves. To accomplish this, we established an ex vivo model, which – in contrast to SC cultures – preserves the 3D architecture and cell-cell-contacts. Such ex vivo culturing of nerves allows for investigation of SC reactions as closely as possible to the in vivo environment in patients. For this, human sural nerves not required as auto-transplant, were freshly collected during surgery. One nerve part was directly frozen

representing the uninjured control condition (0 h). The remaining nerve was cut into smaller pieces and incubated at 37 °C to allow for monitoring of cellular and molecular injury responses (Fig. 1a). Murine sural nerves were treated with exactly the same procedure (Fig. 1a). We included sural nerves from 40 patients (22 males, 18 females) with a median age of 52 years (Supplementary Table 1 and Fig. 1b). Importantly, nerve grafts are devoid of neuronal cell bodies, so responses are attributable to SCs. Indeed, SCs were the major cell type in these biopsies with ~80% of all cells (Supplementary Fig. 1a–c). We observed neither enhanced apoptosis nor cell proliferation within 48 h suggesting that both do not interfere with SC responses observed (Supplementary Fig. 1d–i). Finally, nerve explants lacked infiltrating but also nerve-resident immune cells (Supplementary Fig. 1j–l). This largely excludes immune cells as a source of molecular or cellular responses in this system.

Next, we examined whether SCs in ex vivo cultured nerves trigger typical injury-associated responses described in rodent injury models in vivo. For this, injured nerves were either harvested from living mice or cultured ex vivo followed by gene expression analysis of injury-associated genes (Supplementary Fig. 2). Indeed, mRNA abundance of TFs modulated by nerve injury such as *Egr2*, *cFos*, *cJun*, *Atf3* (activating transcription factor 3) and *Brn2* followed a similar expression pattern in injured nerves in vivo or ex vivo (Supplementary Fig. 2a–d, f). Likewise, genes encoding myelin proteins (*Mbp*, *Pmp22*) or signalling proteins (*Erbb2*, *Gdnf*) showed an almost identical pattern (Supplementary Fig. 2e, g–i). These results suggest that the ex vivo culture system is well-suited to reproduce early injury responses observed in vivo.

We analysed whether Wallerian degeneration was also initiated in cultured nerves. For this, axonal and myelin integrity were histologically analysed at different time points (Fig. 1c–y). Axonal staining revealed reduced axon number over time in both murine and human nerves (Fig. 1c, f, i, l, u, v). In addition, we assessed axonal degeneration in electron microscopy (EM; Supplementary Fig. 3). In line with axonal staining (Fig. 1), fully degraded axons had similar numbers in human and murine nerves, whereas partially degenerated axons were more abundant in murine nerves (Supplementary Fig. 3e).

Besides axonal demise, injured human and murine SCs changed their morphology, partially losing their organised round structure (inserts in Fig. 1d, g, j, m, w, x). Such irregular SC morphologies are typical for Wallerian degeneration, since SCs initiate a dedifferentiation programme after axon severing.

In this acute phase after injury, SCs shed their myelin and phagocyte myelin debris[19]. We used EM to monitor myelin degradation in nerve explants. In mouse nerves, the majority of myelin sheaths degenerated within 24 h after injury (76%; Fig. 1y). This was revealed by impaired integrity, swelling and dissociation of single myelin layers (Fig. 1o, p, y). After 48 h the percentage of degenerated myelin sheaths was slightly higher (81%; Fig. 1q, y), showing a comparable time frame to myelin degeneration described for PNI in vivo[20]. The integrity of myelin sheaths was clearly affected (arrows in Fig. 1p, q), however, only few axons were completely demyelinated (arrowhead in Fig. 1q). Notably, in human explants, myelin sheath degeneration also took place, although significantly decreased compared to murine explants (42% at 24 h and 59% at 48 h; Fig. 1r–t, y).

Thus, axonal degeneration as well as myelin shedding appeared delayed in human nerves.

**Comparison of SC reprograming in human and murine nerves.** A key event of SC reprograming in vivo is the gene expression switch from differentiated to repair SCs[4]. We assessed whether

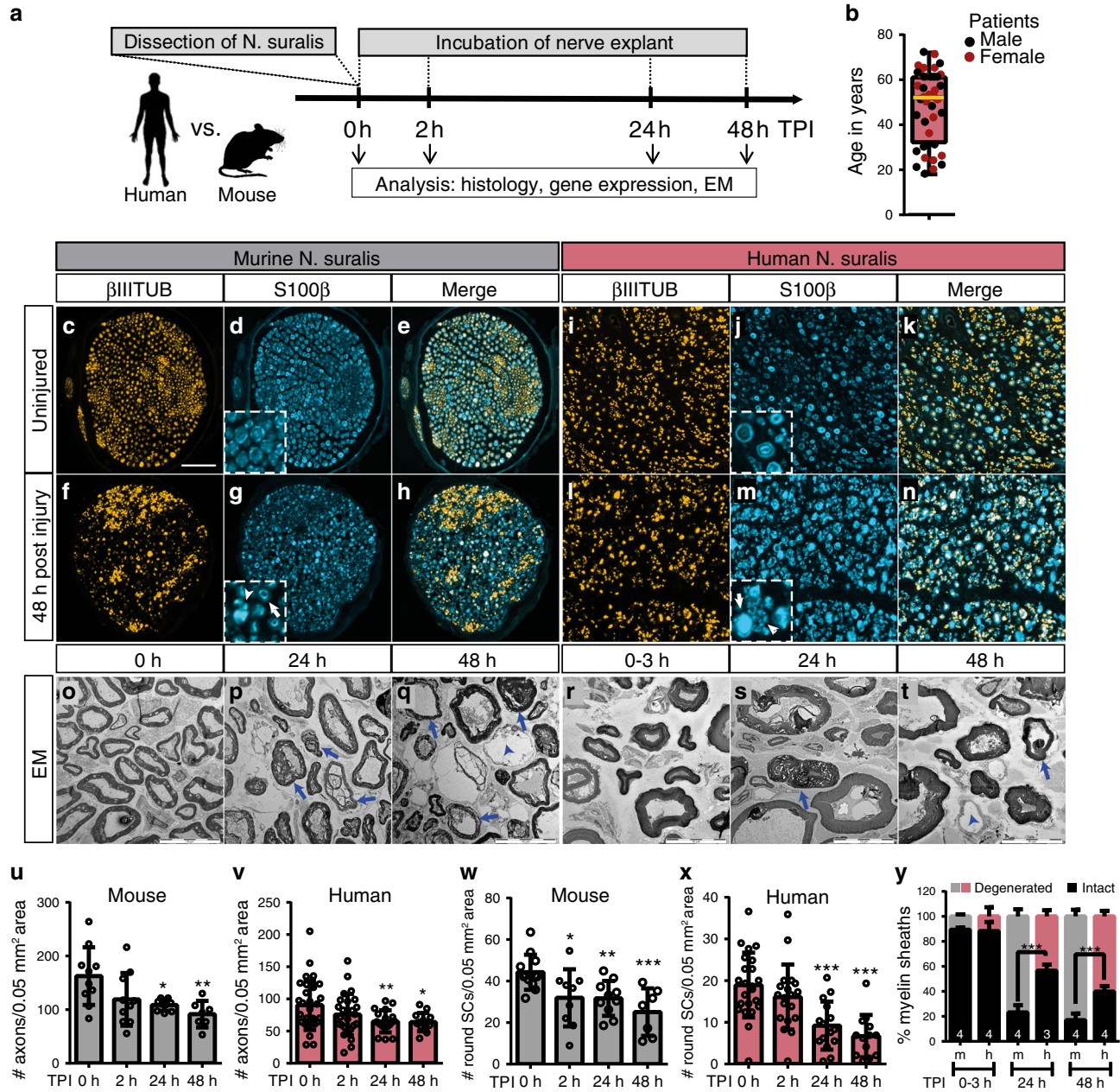

**Fig. 1 Histological analysis of murine and human nerve explants. a** Experimental setup. TPI timepoint post injury, EM electron microscopy. **b** Age and sex distribution of patients. Red and black dots represent female and male patients respectively ($n = 40$ patients; 22 males, 18 females). Box shows 25th to 75th percentiles and median patient age (yellow line), whiskers show min (18 years) to max (72 years). **c–n** Murine (**c–h**) or human (**i–n**) nerves stained for axons (βIIITUB) and SCs (S100β) before or 48 h after injury. Inserts in **d**, **g**, **j**, **m** show higher magnifications. Arrows indicate SCs with round morphology, arrowheads point at SCs with distorted morphology. **o–t** EM pictures of murine (**o–q**) or human (**r–t**) nerves. Arrows (**p**, **q**, **s**, **t**) show degenerated myelin, arrowheads (**q**, **t**) demyelinated axons. **u**, **v** Quantification of axon numbers in murine (**u**) or human (**v**) nerves at different time points. (at 0 h, 2 h, 24 h, 48 h $n = 10, 10, 9$ and 7 biological replicates for murine and $n = 31, 26, 19$ and 13 for human samples respectively). **w**, **x** Quantification of SC numbers in murine (**w**) or human (**x**) nerves at different time points. (at 0 h, 2 h, 24 h and 48 h, $n = 12, 9, 10$ and 7 biological replicates for murine and $n = 23, 19, 15$ and 13 for human samples respectively). Each circle represents a single nerve sample. All error bars show SD. Two-sided Mann–Whitney test was used to calculate statistical significance (*$P < 0.05$, **$P < 0.005$, ***$P < 0.001$). **y** Quantification of intact or degenerated myelin sheaths of EM pictures of human and murine nerves as indicated. Numbers indicate independent biological replicates analysed. Two-sided T-test was used to calculate statistical significance ($P$ value 24 h = 0.0004, $P$ value 48 h = 0.0024). All bars show mean with SD. Statistical significance is shown by asterisks (*$P < 0.05$, **$P < 0.005$, ***$P < 0.001$). Scale bar in **c** is 50 μm and applies for **c–n**. Scale bar in **o** is 10 μm and applies for **o–t**. Source data (**u–y**) are provided as a Source Data file.

this switch is conserved in cultured nerves and – importantly – if differences were obvious when comparing mouse with human nerves. Expression of several TFs labels differentiated SCs (e.g. *Egr2, Brn2*; Fig. 2a, b) whereas other TFs label repair SCs (e.g. *cJun, Atf3*; Fig. 2e, f)[4,21–23].

First of all, gene expression changes described in injured nerves in vivo were reproduced in ex vivo cultured nerves (Fig. 2)[24,25]. This included *cJun, Atf3, Gdnf* and *Shh* induction and *Egr2, Brn2, Pmp22* and *Erbb2* down-regulation (Fig. 2). When inspecting individual genes, first differences in SC reprograming were

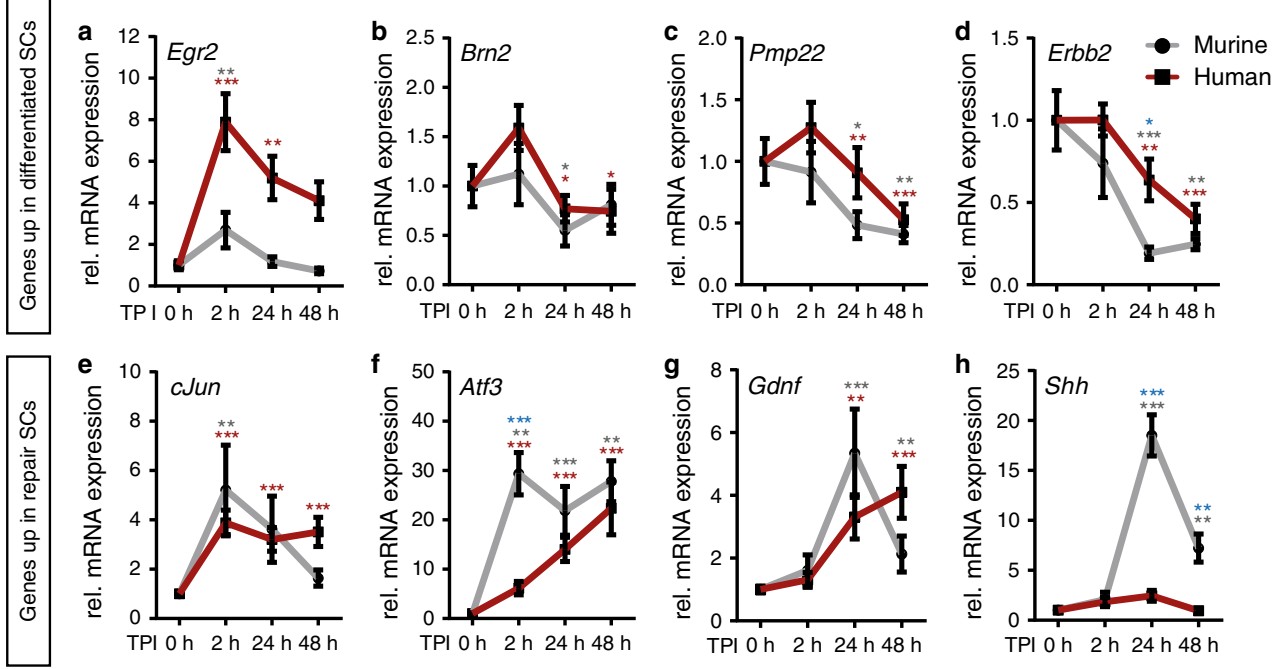

**Fig. 2 Altered gene expression profile in human and murine Schwann cells (SCs) upon injury.** qPCR analysis in human (red line) and murine (grey line) nerve explants at different time point after injury for genes typically expressed in differentiated (**a**–**d**) or repair SCs (**e**–**h**). Expression at 0 h was set to one and the fold change was calculated for the time points post injury (TPI). Graphs show for each time point mean with SEM. Two-sided Mann–Whitney test was used to calculate statistical significance (\*$P < 0.05$, \*\*$P < 0.005$, \*\*\*$P < 0.001$). Grey or red asterisks indicate significance compared to 0 h time point for murine or human samples respectively. Blue asterisks indicate significant differences between mouse and human for the particular time point. Biological replicates: $n$ at 0 h, 2 h, 24 h and 48 h was 7, 7, 7 and 4 for murine nerves respectively, 26, 26, 25 and 14 for human nerves in **a**, **b**, **e** and **f**, and 17, 17, 17, 11 in **c**, **d**, **g** and **h**. Source data are provided as a Source Data file.

observed. For instance, *Egr2* was more abundant in human SCs (Fig. 2a). In contrast, *Atf3*, a regeneration-associated gene[23] expressed in repair SCs, was more abundant in murine nerves (Fig. 2f). Other markers including *Brn2*, *Pmp22*, *Erbb2*, *cJun* and *Gdnf* shared a conserved temporal expression profile in both species (Fig. 2b–e, g). In contrast, *Shh*, a signalling factor up-regulated in repair SCs[5], was induced in murine but not human nerves (Fig. 2h).

In summary, mouse SCs showed a more pronounced repair SC phenotype for selected genes (*Atf3*, *Shh*) compared to human SCs, whereas other genes followed a conserved expression profile.

Since age is an important factor for the extent of nerve regeneration[26,27] we analysed SC reprograming in younger vs. older PNI patients and mice (Supplementary Table 1; Supplementary Fig. 4). Indeed, *cJUN* and *ATF3* were significantly less expressed in older patients 2 h upon injury suggesting reduced repair SC induction (Supplementary Fig. 4a, c). Conversely, *BRN2*, *MBP*, *PMP22* and *ERBB2* were more abundant in older PNI patients (Supplementary Fig. 4e, g, i, k). This scheme was conserved when comparing younger (2 months) vs. older (6 months) mice (Supplementary Fig. 4b, d, f, h, j, l).

Thus, selected SC reprograming genes reveal an expression profile matching the regeneration potential of injured nerves in younger vs. older PNI patients.

**Genome-wide transcriptomics in human vs. mouse SCs.** The first differences observed in human vs. moue SC reprograming (Figs. 1 and 2) prompted us to perform genome-wide transcriptomics. Therefore, ex vivo incubated murine and human nerves were subjected to microarray analysis at 0 h, 2 h and 24 h after injury ($n = 3$ mouse or 5 human nerves for each time point). In general, we focused on gene encoding mRNAs (Figs. 3 and 4)

but differences and similarities in non-coding RNAs were observed (Supplementary Data Set 1).

Surprisingly, when comparing human nerves before and 2 h after injury, no genes were significantly and ≥2.0-fold up- or down-regulated (Fig. 3a). In contrast, in murine nerves, 25 and 35 genes were down- or up-regulated respectively 2 h after injury (Fig. 3b). Most up-regulated genes were IEGs (e.g. *Egr1*, *cFos*, and *cJun*) whose induction was still present 24 h after injury in murine nerves (Fig. 3c). Furthermore, *Atf3* (Fig. 3c) and *Shh* (Supplementary Dataset 1) were up-regulated in mouse but not human nerves, thus corroborating our quantitative polymerase chain reaction (qPCR) analysis (Fig. 2). In general, IEG induction was modest at 2 h in human nerves and somewhat stronger at 24 h after injury (Fig. 3c).

At 24 h after injury, both human and murine nerves up- or down-regulated more genes (Fig. 3d, e) compared to 2 h (Fig. 3a, b). Still, more than twice the number of genes were ≥2.0-fold altered in mice compared to human nerves (mouse: 952 genes; human: 412 genes; Fig. 3d, e). In both species, an up-regulated gene set was associated with inflammation as evident by GO term analysis (Fig. 3f). Previously, SCs were reported to secret several cytokines and chemokines[19]. In agreement, in ex vivo incubated murine and human nerves, numerous genes related to the immune system including many CCL and CXCL chemokines were up-regulated (Fig. 3g). We confirmed this inflammation-related gene induction using qPCR (Supplementary Fig. 5). Induction of inflammatory genes was almost identical in human and mouse nerves (Fig. 3; Supplementary Fig. 5) pointing at a species-conserved injury response in line with the literature[9]. Since immune cells are essentially absent in our nerve preparations (Supplementary Fig. 1), SCs were likely the source for chemokine and interleukin production.

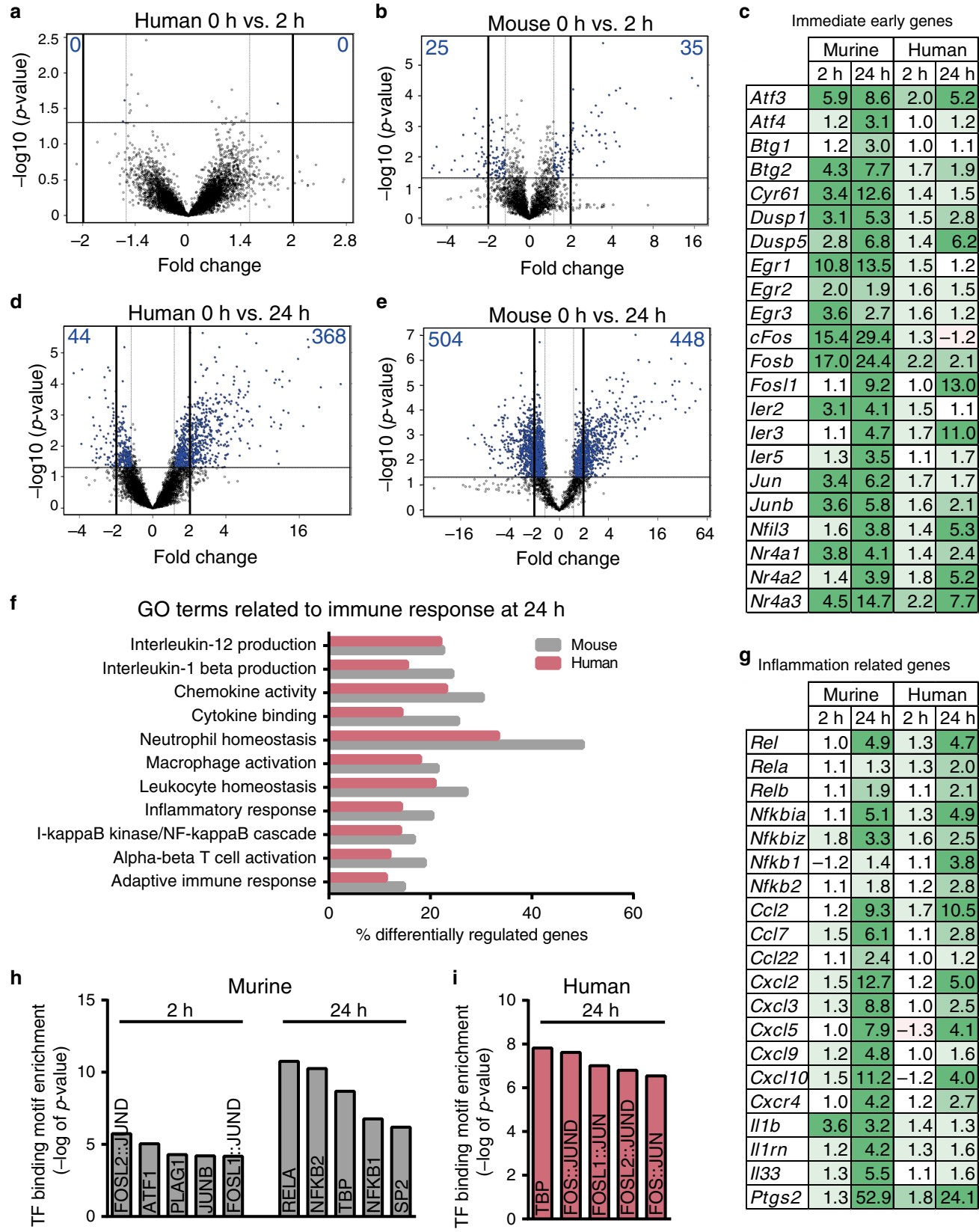

TF binding motif analysis in mice identified JUN and FOS family members 2 h after injury (Fig. 3h), a finding in accordance with IEG induction (Fig. 3c). At 24 h, the predominant response in murine nerves was related to NF-κB activity, fitting with immune gene induction (Fig. 3h). In opposite, in human nerves the major TF binding motif was JUN/FOS member associated 24 h after injury (Fig. 3i), correlating with the delayed IEG induction at this time point (Fig. 3c).

In summary, human and mouse nerves share an inflammatory gene response but differ in IEG induction.

**Fig. 3 Transcriptomic analysis of human vs. murine nerve explants. a, b** Volcano plots of differentially regulated genes in human (**a**) or murine nerves (**b**) 2 h upon injury compared to uninjured nerves. Blue numbers indicate the number of more than two-fold down (left side) or up (right side) regulated genes. **c** Fold change expression of selected IEGs in murine/human nerve explants at 2 h and 24 h after injury compared to 0 h. **d, e** Volcano plots of differentially regulated genes in human (**d**) or murine nerves (**e**) 24 h upon injury compared to uninjured nerves. **f** GO (Gene ontology) terms related to immune responses were significantly altered in human and murine nerves 24 h upon injury. Source data are provided as a Source Data file. **g** Fold change expression of selected inflammation-related genes in murine/human nerves at 2 h and 24 h after injury compared to 0 h. Fold change in **c** and **g** was calculated from the mean normalised intensities irrespective of significance. Significantly changed values are depicted in bold. **h, i** TF binding motif enrichment analysis for genes significantly up-regulated at least threefold in murine nerves 2 h and 24 h (**h**) and for human nerves 24 h (**i**) after injury. P-values were calculated by Pscan Ver. 1.5 software using a two-tailed Z-test.Analysed biological replicates in **a–i**: human n = 5, murine n = 3 for each time point.

**Murine but not human SCs adapt lipid metabolism upon injury.** One striking injury-induced change in murine SCs was adaption in metabolism affecting glycolysis, citric acid cycle and most obviously lipid metabolism (Fig. 4, S6 and S7). Particularly adaptations in lipid metabolism appear reasonable since repair SCs discontinue the energetically expensive myelin production (Fig. 1)[19]. Indeed, many GO terms associated with lipid metabolism were altered in murine but essentially absent in human nerves (Fig. 4a, Supplementary Fig. 7). Closer inspection revealed that in murine nerves more than 50 lipid metabolism encoding genes (referred to as lipogenic genes) were two-fold or more down-regulated whereas this was less pronounced in human nerves (Fig. 4b). Down-regulated genes included TFs such as *Pparg* and *Srebp1* involved in lipid gene regulation[28,29]. In addition, key regulatory enzymes of fatty acid synthesis including *Acsl1* (acyl-Coenzyme A synthetase), *Acaca* (acetyl-Coenzyme A carboxylase), *Fasn* (fatty acid synthase) and *Dgat2* (diacylglycerol O-acyltransferase 2; Fig. 4b) were down-regulated after injury. Further genes encoded proteins for lipid β oxidation (*Echs1*, enoyl-CoA hydratase short chain 1; *Ehhadh*, enoyl-CoA hydratase and 3-hydroxyacyl CoA dehydrogenase; Fig. 4b), transport and storage (e.g. *Plin1*, perilipin 1; *Cidec*, cell death inducing DFFA like effector c; *Lpl*, lipoprotein lipase; Fig. 4b and S6).

qPCR analysis in independent nerve samples confirmed this mRNA down-regulation of lipogenic genes in murine and the weak down-regulation in human nerves (Fig. 4c–j). We investigated whether the reduction in lipogenic gene expression in human nerves is generally blunted or delayed. In human nerves incubated up to 5 days, the abundance of several genes of lipid anabolism (SREBP, ACACA, ACSL1, FASN and DGAT2) also dropped down to the low mRNA levels obtained in mice already at two days post injury (Supplementary Fig. 8). In contrast, genes related to lipid catabolism (ECHS1, EHHADH) were induced 3–5 days after injury. These results indicate a delayed adaptation of lipid metabolism in human nerves.

Beside qPCR analysis, PPARg protein down-regulation in SCs was confirmed in teased murine sural nerve fibres at 24 h post injury (Fig. 4k, l). Of note, down-regulation of lipogenic genes was confirmed in injured murine nerves harvested in vivo (Fig. 4m) thereby documenting that ex vivo cultured nerves respond similarly to nerves injured in vivo.

Differentially regulated genes included two TFs previously described as regulators of lipid metabolism in non-neuronal tissues, *Pparg* (and its co-factor retinoic X receptor g; *Rxrg*) and *Srebp1* (Fig. 4b-d)[28–30]. TF binding motif enrichment analysis of down-regulated genes in murine nerves, uncovered PPARg together with RXR as candidates responsible for regulation of those lipogenic genes (Fig. 4n) in agreement with microarray and qPCR data (Fig. 4b, c). In contrast, these TFs did not emerge in the TF binding motif analysis of human nerves, in line with the decreased lipid metabolism shut-down in human SCs at this time point (Fig. 4o). In contrast to repair vs. differentiated SC markers (Supplementary Fig. 4), lipogenic gene expression was not age dependent (Supplementary Fig. 9).

Investigation of lipid metabolism identified a first gene set regulated in mouse but only delayed in human nerves (Fig. 4). Interestingly, we also observed genes regulated in human but not murine nerves. One such gene strongly up-regulated (~10-fold) in human nerves was *MEDAG* (mesenteric estrogen dependent adipogenesis), not altered in murine nerves (Fig. 4b). qPCR validation confirmed *MEDAG* induction in injured human nerves while in mouse nerves *Medag* expression was even reduced (Fig. 4p). *Medag* has previously been associated with *Pparg* and lipogenic gene expression in adipose tissue[31]. Hence, this so far poorly characterised gene might be a novel candidate for cross-species differences between human and murine injured nerves.

**Reduced sphingosine-1-phosphate levels in injured murine SCs.** For deciphering effects of lipid gene regulation directly on lipid level, lipidomics were performed. Since mRNA alterations in fatty acid synthetising and degrading enzymes were observed (Fig. 4; Supplementary Fig. 6) one might expect changes in overall fatty acid abundance affecting structural and signalling lipids. Therefore, we analysed levels of structural membrane lipids such as glycerolipids (e.g. phosphatidylcholine (PC), phosphatidylethanolamine (PE)) and sphingolipids (sphingomyelin (SM)). Furthermore, glycerophospho- and sphingolipid species with established signalling properties were tested. The latter included lysophosphatidic acid (LPA), different phosphatidylinositols (PI), phosphatidylserine (PS) and sphingolipids such as ceramide (CER), sphingosine (SPH) and sphingosine-1-phosphate (S1P). Those sphingolipids have well-known functions as messenger molecules, regulating processes like survival, proliferation, differentiation, migration, cytokine secretion and inflammation[32–35]. In addition, CER was previously shown to induce myelinophagy[19,33].

We analysed 5 human and 5 murine nerves 0 h and 24 h after ex vivo incubation. Lipidomic analysis revealed that injured murine and human nerves differed in their lipid profile (Fig. 5a). In mice, uninjured and injured samples were found in separate clusters suggesting specific alterations in mouse lipidome after injury. In contrast, uninjured and injured human nerves were randomly clustered suggesting no consistent lipidome alterations (Fig. 5a). Thus, lipidome analysis is congruent with microarray results (Fig. 4) suggesting lipid adaptions in mouse but only weakly in human nerves after injury.

In total, abundance of 27 lipids was analysed (Fig. 5a). Out of those, 12 (44%) were significantly changed in injured mouse nerves whereas only 4 (15%) were modulated in human nerves (Fig. 5a, b). Intriguingly, among the most prominently altered lipids in murine nerves we found sphingolipids like CER, SPH and S1P. Our data suggest increased abundance of CER and SPH at the expense of S1P whose abundance decreased after injury (Fig. 5b). Beside sphingolipids we found several glycerolipids including lysophosphatidylcholine (LPC), lysophosphatidylinositol (LPI), LPA, phosphatidylglycerol (PG), PE and specific PIs almost exclusively regulated in injured mouse nerves (Fig. 5b). In

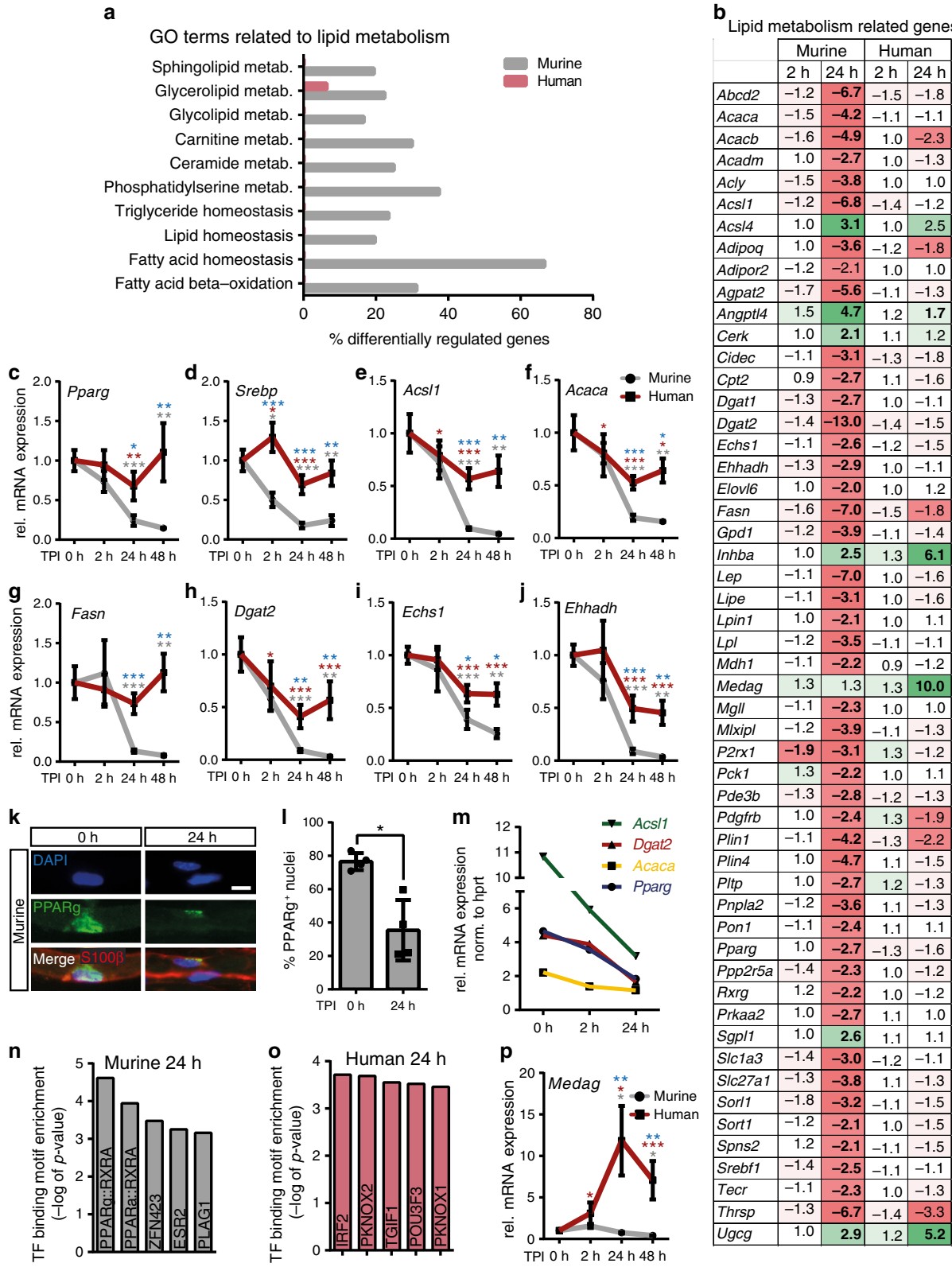

contrast, abundance of selected poly unsaturated fatty acids (PUFAs) was not obviously changed by nerve injury (Supplementary Fig. 10).

Taken together, our data suggest that sphingolipid signalling may be a key component in regulation of acute responses in injured murine but not as much in human nerves.

**S1P and PPARg regulate lipogenic gene expression**. Increased S1P signalling through S1P receptor (S1PR) induces a proregenerative repair SC phenotype[36]. Lipidomics revealed decreased S1P levels in injured murine nerves (Fig. 5). This was in line with the increased *Sgpl1* expression after injury, an enzyme degrading S1P into phosphoethanolamine and hexadecanal

**Fig. 4 Down-regulation of lipid metabolism in injured murine but not human nerves. a** Alterations in GO (Gene Ontology) terms in murine/human nerves 24 h upon injury. **b** Fold change expression of lipid metabolism related genes in murine/human nerves after injury compared to 0 h. Significantly changed values are depicted in bold. **c–j** qPCR validation of selected lipogenic genes at different time points post injury (TPI). Bars show mean with SEM. Grey or red asterisks indicate significance compared to 0 h time point for murine or human samples respectively. Blue asterisks indicate significant differences between mouse and human for the particular time point. **k**, **l** Down-regulation of PPARg+ in SCs of teased nerve fibres. Scale bar is 10 μm. Bars in **l** show mean with SD ($P = 0.0286$). **m** qPCR of selected lipogenic genes in nerves injured in vivo. Graph shows mean with SEM for each time point. **n**, **o** Transcription factor (TF) binding motif analysis for genes significantly down-regulated at least threefold in murine (**n**) or human (**o**) nerves 24 h after injury. *P*-values were calculated by Pscan Ver. 1.5 software using a two-tailed *Z*-test. **p** qPCR validation of expression of the gene *Medag* in human/murine nerves at different time points post injury. Analysed biological replicates: for (**a**, **b**, **n–o**) human $n = 5$, murine $n = 3$ for each time point, for (**c–h**) $n = 15, 15, 15$ and 12 for human and 10, 10, 10 and 4 for mouse at 0 h, 2 h, 24 h and 48 h respectively, for (**i**, **j**, **p**) $n = 7, 7, 7$ and 7 for human and 10, 10, 10 and 4 for mouse at 0 h, 2 h, 24 h and 48 h respectively, for **l** $n = 4$ for each time point, for **m** $n = 9, 3$ and 3 at 0 h, 2 h and 24 h respectively. Two-sided Mann–Whitney test was used to calculate statistical significance (*$P < 0.05$, **$P < 0.005$, ***$P < 0.001$). Source data are provided as a Source Data file.

(Fig. 4b). So far, the impact of functional modulation of S1P and PPARg activity was not analysed in injured SCs. However, previous studies showed S1P interaction with PPARg to upregulate lipogenic genes[37,38]. In addition, pharmacological PPARg activation rescues developmental myelination defects in SCs[39].

Next, we investigated whether changes in S1P level and PPARg activity affect lipogenic gene expression in injured nerves (Fig. 6). To modulate S1P signalling, murine nerves were treated with the SGPL1 inhibitor 4-deoxypyridoxine (DOP) shown to elevate S1P levels[40–43]. As before (Fig. 4), genes related to lipid metabolism underwent an injury-induced down-regulation in control-treated nerves (Fig. 6a). Notably, DOP strongly up-regulated expression levels of several lipogenic genes including *Pparg* itself, *Srebp*, *Acaca*, *Fasn* and *Dgat2* to at least pre-injury levels, sometimes even exceeding the uninjured control condition (Fig. 6a). DOP is a frequently used SGPL1 inhibitor[40–43], however since it functions as a vitamin $B_6$ antimetabolite, other vitamin $B_6$ depending enzymes might also be targeted. To further elaborate the role of S1P in regulating lipogenic gene expression, we used two additional SGPL1 inhibitors, 2-acetyl-5-tetrahydroxybutyl imidazole (THI) and compound 31 (C31)[44–47]. Both inhibitors showed similar effects as DOP, although to a somewhat lower extent (Supplementary Fig. 11)

To provide further support for a role of PPARg in PNI associated gene expression murine nerves were treated with the PPARg agonist pioglitazone (PIO; Fig. 6b). Since PPARg transcriptionally activates several lipid metabolism encoding genes[29] we expected induction of those genes by PIO application. Similar to DOP, PIO treatment induced lipogenic genes after injury, now reaching almost pre-injury levels for many genes (Fig. 6b). These results point at a role of PPARg function during nerve injury-induced down-regulation of lipogenic genes.

Besides lipogenic genes (Fig. 6a, b), we investigated the impact of DOP and PIO on marker genes for repair (*Shh*, *Gdnf*) or differentiated SCs (*Mbp*; Fig. 6c–h). Notably, both DOP and PIO suppressed induction of *Shh* and *Gdnf* expression after injury, while PIO additionally induced *Mbp*, thereby apparently favouring the myelinating SC over the repair SC phenotype. The influence of PIO on SC reprograming was also tested on protein level (Fig. 6i–n). For this, the abundance of cJUN – the prototypical TF present in repair SCs[4] – was analysed in murine nerve explants. Expectedly, cJUN up-regulation was observed after injury in SCs (Fig. 6i, l). In contrast, PIO-mediated PPARg activation impeded cJUN up-regulation (Fig. 6i, l). cJUN was localised to DAPI+ nuclei in injured nerves as expected (Supplementary Fig. 12a, left side). Total number of DAPI+ nuclei was unaltered by PIO thereby excluding reduction of cJUN + cells simply by cell loss (Supplementary Fig. 12b). Besides, we analysed MBP as SC differentiation marker. Expectedly, MBP levels were reduced after injury due to myelin degradation, while PIO treatment preserved MBP levels in injured nerves (Fig. 6j, m).

Finally, we investigated whether PPARg signalling affects axonal degradation after ex vivo nerve injury. In control-treated murine nerves, axon clearance was observed after injury (Fig. 6k, n) as before (Fig. 1). In contrast, in PIO-treated nerves axon clearance was diminished (Fig. 6k, n). Further, EM was used to evaluate axonal degradation (Supplementary Fig. 13). Here, we likewise observed decreased debris clearance within the axonal compartment in PIO-treated injured nerves (Supplementary Fig. 13).

Taken together, our data indicate that PPARg activation interferes with SC reprograming by favouring SCs to remain in a differentiated rather than entering a repair state. Hence, PPARg may play an important role in transition between myelinating and repair SCs.

**Human SCs respond to pharmacological PPARg modulation.** Since PIO is an FDA approved anti-diabetic drug with beneficial effects on the lipid profile in patients[48–50], we tested its impact on human SCs (Fig. 7). As described above, in human nerves down-regulation of lipid metabolism was less pronounced than in murine nerves (Fig. 4a–j) thus a weaker PIO impact might have been anticipated. Intriguingly, similarly to murine nerves (Fig. 6b), PIO increased expression of genes like *PPARg, ACACA, FASN* and *DGAT2* in human nerve samples (Fig. 7a). In fact, pharmacological PPARg activation even resulted in higher mRNA abundance than in uninjured nerves, which was probably due to the limited *PPARg* down-regulation after injury.

To corroborate findings on PPARg's role in SC reprogramming in human tissue, we assessed cJUN expression without or after PIO treatment. Similar to murine nerves, human nerves upregulated cJUN after injury (Fig. 7b, e). Of note, cJUN positive SCs were already observed at the 0 h time point, which is likely due to the fact that dissected human nerves in this experiment had a delay until being frozen during surgery (Methods section). Nevertheless, also here pioglitazone treatment abolished cJUN expression (Fig. 7b, e). Nuclear cJUN localisation and unaltered nucleus numbers were also confirmed in human nerves (Supplementary Fig 12a right side, c). Moreover, similar to murine nerves, MBP decreased in injured human nerves, while PIO impeded on this decrease (Fig. 7c, f). Finally, we investigated whether axonal degradation was also altered by PIO as observed for murine nerves. In control-treated human nerves axon clearance was observed after injury (Fig. 7d, g), although the rate seemed to be lower than in murine nerves (Fig. 6n). Yet, in PIO treated nerves axon clearance was slightly diminished (Fig. 7d, g).

In a final step, we pharmacologically blocked PPARg activity, which should decrease lipogenic gene expression and thereby presumably promote SC reprogramming. For this, injured human nerves were treated with the PPARg antagonists SR16832 (SR) and GW9662 (GW) and analysed at 48 h after injury[51–54].

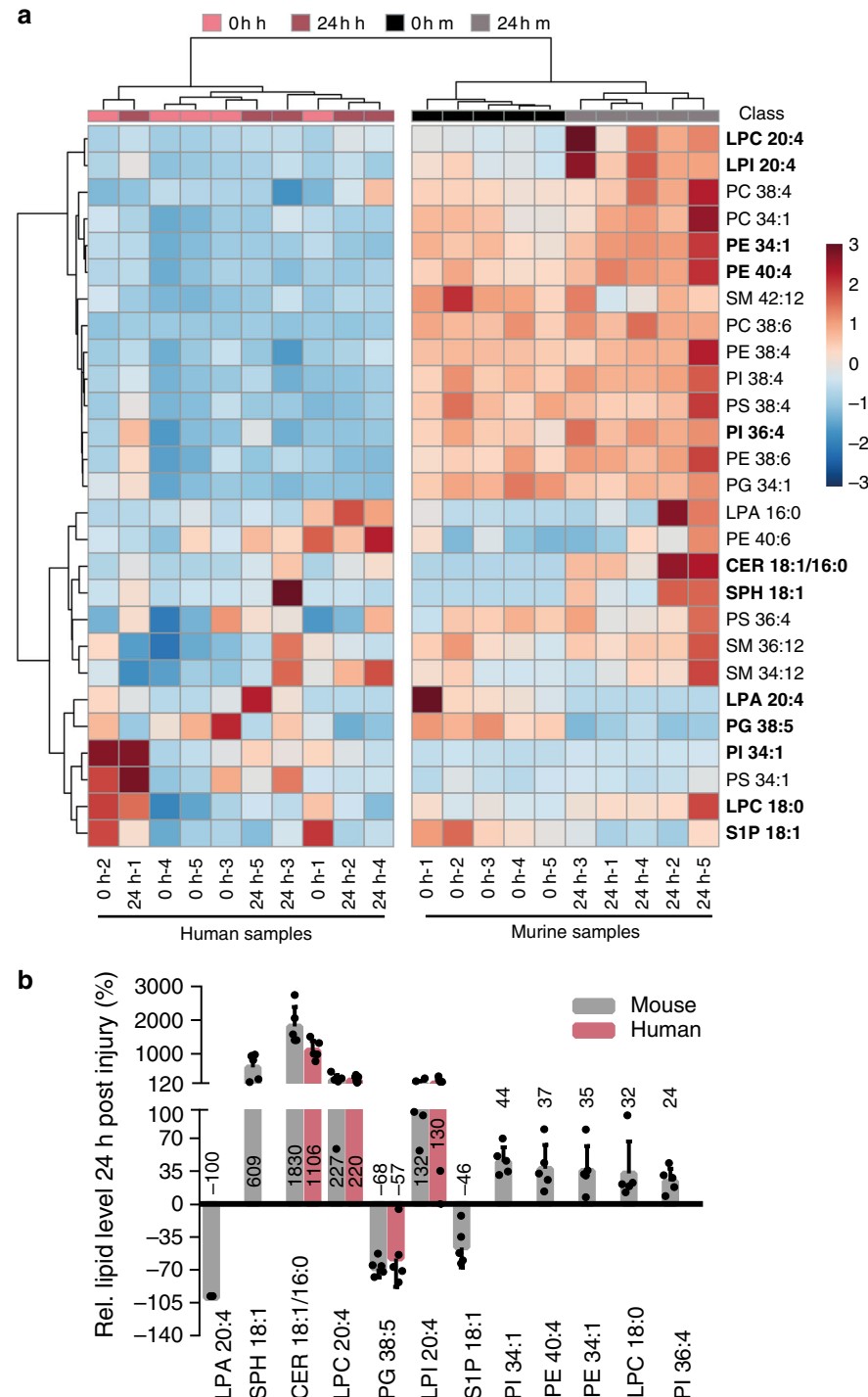

**Fig. 5 Lipidomic analysis in human and murine injured nerves. a** Heatmap of all analysed lipids in human (h) and murine (m) nerves without (0 h) or 24 h after injury. Significantly changed lipids at 24 h are indicated in bold and separately depicted in **b**. **b** Relative lipid level for all significantly changed lipids upon injury in human or murine nerves. For each lipid the mean level of all human or murine samples at 0 h was set to 100% and the change was calculated at 24 h after injury. $n = 5$ biological replicates for each time point for human and murine samples. Each dot represents a single murine or human sample analysed. Bars show mean with SD. Source data are provided as a Source Data file.

Indeed, both PPARg antagonists resulted in further down-regulation of lipogenic genes to half the expression levels observed of untreated injured nerves (Fig. 7h). This suggests that regeneration in human nerves at an early injury stage, when SCs still have to reprogramme, can be modulated by PPARg antagonists.

Overall, PPARg plays a role in regulating gene expression of enzymes involved in fatty acid metabolism in murine and human nerves and its down-regulation after injury appears to be important for the adaptation of lipid metabolism during SC reprograming (see graphical summary in Fig. 8).

## Discussion

Herein we established an ex vivo model for investigation of SC adaptations in injured peripheral nerves. This system monitors

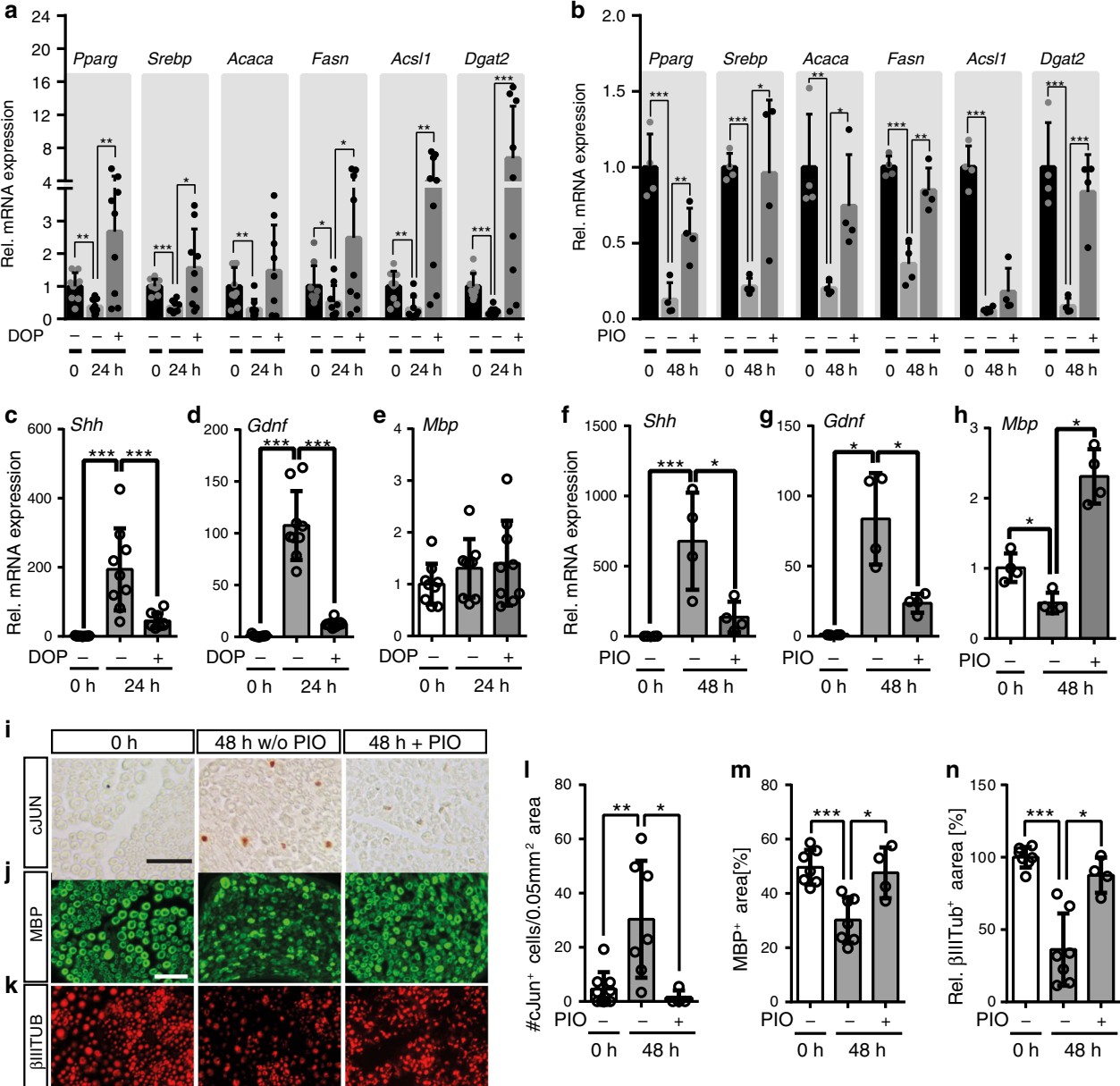

**Fig. 6 S1P/PPARg dependent regulation of lipid metabolism and SC reprogramming in mice. a**, **b** qPCR analysis of genes involved in lipid metabolism in control or injured murine nerves treated with 4-deoxypyridoxine (DOP; **a**) or pioglitazone (PIO; **b**). Expression at 0 h was set to one and the fold change was calculated for the other time points. Biological replicates analysed: $n = 9$ and $n = 4$ for each condition for (**a**) and (**b**) respectively. **c–h** qPCR analysis of the marker genes for repair SCs *Shh* and *Gdnf* and the myelin gene *Mbp* in control or injured murine nerves treated with DOP (**c–e**) or PIO (**f–h**). Expression at 0 h was set to one and the fold change was calculated for the other time points. Biological replicates analysed: $n = 9$ for **c–e** and $n = 4$ for **f–h** for each condition. **i–k** Histological analysis of cJUN protein expression (**i**), MBP (**j**) and βIII tubulin+ axons (**k**) in murine nerves without injury (0 h) or 48 h after injury with control (w/o PIO) or with pioglitazone (+ PIO) treatment. Scale is 25 μm. **l** Quantification of cJUN+ cells per nerve area. Biological replicates: $n = 9$, 7 and 4 for 0 h, 48 h and 48 h + PIO. **m** Quantification of MBP+ area per nerve area. Biological replicates: $n = 7$, 7 and 4 for 0 h, 48 h and 48 h + PIO. **n** Quantification of the relative βIIITUB+ area per nerve area. Time point 0 h was set to 100%. Biological replicates: $n = 7$, 7 and 4 for 0 h, 48 h and 48 h + PIO. Each dot represents a single mouse nerve. Bars in all graphs show mean with SD. Two-sided Mann–Whitney test was used to calculate statistical significance (*$P < 0.05$, **$P < 0.005$, ***$P < 0.001$). Source data for **a–h**, **l–m** are provided as a Source Data file.

human and mouse SC responses under in vivo like conditions at earliest post PNI stages (up to 5 days after injury). We noted reduced repair SC gene induction in injured human compared to murine nerves. For instance, differentiated SC markers (e.g. *Erbb2*, *Egr2*) showed a more efficient down-regulation in murine nerves (Fig. 2). Conversely, repair genes (e.g. *Shh*, *Atf3*) were induced faster/stronger in murine than human nerves (Fig. 2). Furthermore, Wallerian degeneration appeared accelerated in murine nerves (Fig. 1). Together, this supports the presumption

of a faster transition of differentiated into repair SCs in murine nerves (Fig. 8a). Of note, first data provided on human nerves suggest that age is a further factor affecting regeneration outcome. Repair gene induction was more pronounced in younger compared to nerve samples of older patients (Supplementary Fig. 4) indicating that nerves of younger patients more readily induce the repair SC programme.

We performed transcriptomics to obtain a comprehensive view of all injury-associated processes. Here, profound changes

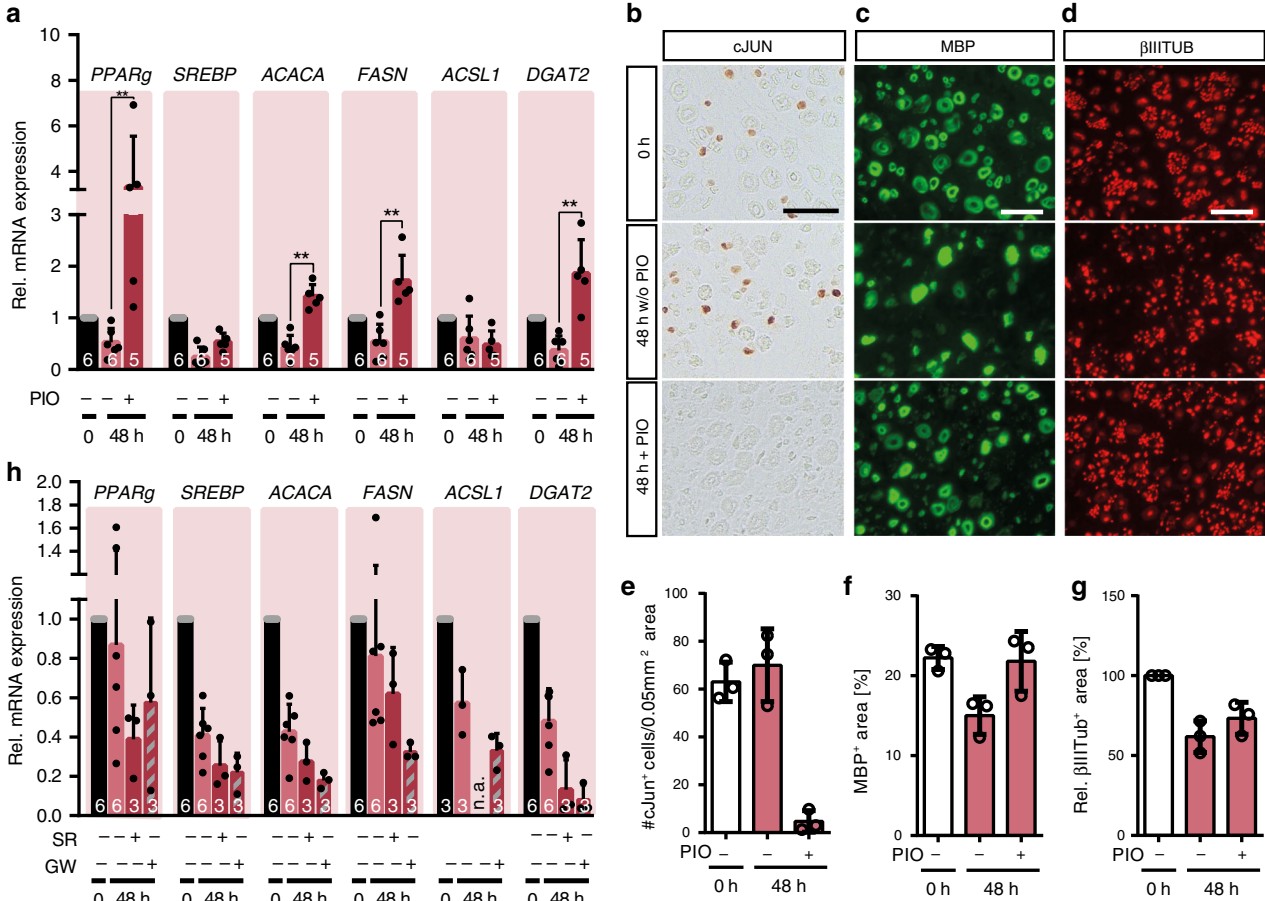

**Fig. 7 Human SCs are responsive to pharmacological PPARg modulation. a** qPCR analysis of genes involved in lipid metabolism in control or injured human nerves treated with pioglitazone (PIO). Expression at 0 h was set to one and the fold change was calculated for the other time points. Numbers in bars indicate independent samples analysed. *P* value for 48 h vs. 48 h + PIO = 0.0043 each for *Pparg*, *Acaca*, *Fasn* and *Dgat2*. **b–d** Histological analysis of cJUN protein expression (**b**), MBP$^+$ area (**c**) and βIIITUB$^+$ axons (**d**) in human nerves without injury (0 h) or 48 h after injury with control (w/o PIO) or with pioglitazone (+ PIO) treatment. Scale: 25 μm. **e–g** Quantification of cJUN$^+$ cells per nerve area (**e**), MBP$^+$ area per nerve area (**f**) and relative βIIITUB$^+$ area per nerve area (**g**). Time point 0 h was set to 100% for (**g**). **h** qPCR analysis of genes involved in lipid metabolism in control or injured human nerves treated with the PPARg antagonists SR16832 (SR) and GW9662 (GW). Expression at 0 h was set to one and the fold change was calculated for the other time points. Numbers in bars indicate independent samples analysed (n.a., not available). Each dot or circle represents a single human nerve analysed. Biological replicates analysed: for **a** n = 6, 6 and 5 for 0 h, 48 h and 48 h + PIO respectively, for **e–g** n = 3 for each condition, for **h** n = 6, 6, 3 and 3, 0 h, 48 h, 48 h + GW and 48 h + SR respectively. Bars in all graphs show mean with SD. Two-sided Mann–Whitney test was used to calculate statistical significance (*P < 0.05, **P < 0.005, ***P < 0.001). Source data for **a**, **e–h** are provided as a Source Data file.

particularly in lipid metabolism were observed in rodent but not as much in injured human nerves (Fig. 4; Supplementary Fig. 7). The importance of lipid metabolism during nerve development and myelin maintenance is widely acknowledged[39,55,56] and changes in lipid content, metabolism and storage after nerve injury were previously observed[57–59]. However, detailed knowledge on changes of lipid metabolism during SC reprograming in PNI is largely missing.

The metabolic adaptation in injured mouse nerves comprised down-regulation of a gene set encoding enzymes of lipid synthesis (Fig. 4, Supplementary Fig. 6). This down-regulation of lipogenic enzymes such as the rate-limiting *Acaca* producing malonyl-CoA and the major fatty acid producing enzyme *Fasn* might stop de novo production and facilitate removal of lipid-rich myelin during Wallerian degeneration in mice. The key role of *Fasn* for myelin production in SCs was recently demonstrated in *Fasn* mouse mutants resulting in impaired myelin membrane production during development[39].

When comparing microarray and lipidomic data, changes in mRNA abundance for lipogenic enzymes were in part matched by

lipid levels in injured murine nerves (Figs. 4b and 5). The combination of transcriptomics and lipidomics suggests S1P/PPARg signalling as potential regulatory unit mediating adaptation of lipogenic gene expression during mouse nerve injury. *Pparg* mRNA and protein were diminished in injured murine but not as strongly in human nerves (Fig. 4). Likewise, lipidomics revealed decreased S1P abundance in injured murine nerves whereas other sphingolipids (CER, SPH) were up-regulated. Microarray data suggest that such decreased S1P levels might be achieved through concomitant transcriptional up-regulation of *Sgpl1*, the major S1P degrading enzyme (Figs. 4b and 8). Notably, outside the nervous system S1P is described as a potent PPARg activator and vice versa, PPARg regulates S1P levels[37,38,60]. Thus, our data suggest that S1P/PPARg might likewise form such a functional unit in nerves.

How might S1P/PPARg signalling be associated with SC reprogramming during PNI?

After injury, SCs adopt a repair phenotype. Our data suggest that acquisition of this repair state in mice is accompanied by adaptions in the SC lipid metabolism on transcriptome and

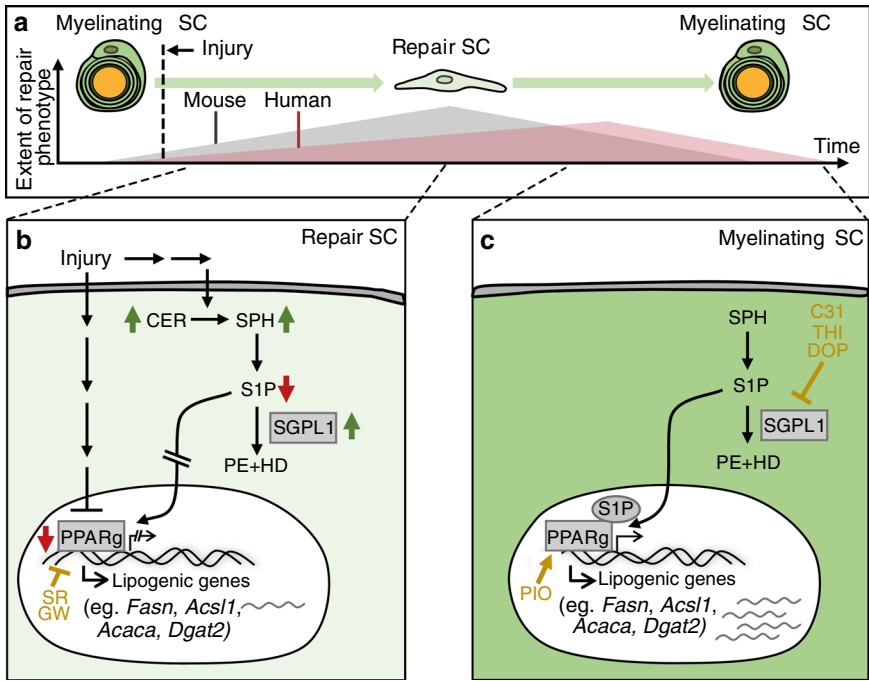

**Fig. 8 Proposed mechanism for lipid metabolism regulation in SCs upon injury. a** After injury, SCs switch from a myelinating (dark green) to a repair (light green) phenotype. In human SCs (red area), induction of the repair SC phenotype is decreased or at least delayed, which might lead to impaired regeneration and delayed re-differentiation compared to murine SCs (grey area). **b** In murine repair SCs, PPARg expression is down-regulated after injury. In addition, intracellular S1P is decreased which might involve enhanced enzymatic degradation to PE and HD by the enzyme SGPL1. Both, decreased PPARg and S1P levels result in blunted PPARg activity, which in turn leads to decreased expression of lipid metabolism associated genes. Pharmacological repression of PPARg activity in human SCs by SR and GW decreased lipogenic gene transcription. **c** Later during regeneration SCs re-differentiate into myelinating SCs. In this case PPARg expression and S1P levels have to be raised again, thereby enhancing PPARg activity and lipogenic gene expression. Pharmacological elevation of S1P by DOP-, THI-, C31-mediated SGPL1 inhibition, as well as PIO-mediated PPARg activation were able elevate lipogenic gene transcription and led to suppression of the repair SC phenotype. C31 (compound 31), DOP (4-deoxypyridoxine), GW (GW9662), HD (hexadecanal), PE (phosphoethanolamine), PIO (pioglitazone), PPARg (peroxisome proliferator-activated receptor gamma), S1P (sphingosine-1 phosphate), SC (Schwann cells), SGPL1 (sphingosine-1 phosphate lyase 1), SR (SR16832) and THI (2-acetyl-5-tetrahydroxybutyl imidazole).

lipidome level. This adaption appears specific to injured murine nerves and is delayed in human nerves (Fig. 8, Supplementary Fig. 8). Our model suggests that in mice, decreased abundance of S1P/PPARg would stall lipid production and thereby facilitate myelin removal and induction of SC repair phenotype (Fig. 8b). This scenario is supported by the experimental finding that pharmacological up-regulation of S1P/PPARg activity enhanced lipogenic genes in injured mouse nerves (Fig. 6), while PPARg inactivation decreased their expression (Fig. 7). Furthermore, PIO and DOP interfered with induction of SC repair markers (*Gdnf*, *Shh* and cJUN; Figs. 6 and 7). Thus, S1P/PPARg inhibition appears necessary for initiating the SC switch towards repair SCs after PNI. Of note, as nerve regeneration proceeds, regenerating axons once again have to be myelinated therefore requiring a switch of repair SCs back to myelinating SCs (Fig. 8c). Data provided herein suggest that PIO, DOP, THI or C31 stimulate this transition and enhance adoption of a myelinating SC fate through S1P/PPARg mediated expression of lipogenic genes (Figs. 6, 7 and 8c, Supplementary Fig. 11). Thus, although speculative, individual or combined application of PIO, DOP, THI or C31 during later regeneration stages might be beneficial for the final re-myelinating phase required during axonal regeneration. Accordingly, PIO improves remyelination of injured nerves in vivo. However, this was mediated by macrophages, while a direct SC involvement – as suggested by our data – was not investigated[61].

Interestingly, although human SCs were largely refractory to changes in lipid metabolism (Figs. 4 and 5), they reacted to

pharmacological PPARg activation with enhanced induction of selected lipogenic genes (Fig. 7a). Conversely, pharmacological PPARg repression resulted in down-regulation of several lipogenic genes (Fig. 7h). This suggests that human SCs are in principle responsive to pharmacological modulation of PPARg activity, a finding which might have some translational impact in human PNI.

Currently, it is unknown why in human SCs initiation of a signalling pathway resulting in altered lipid metabolism is delayed. Herein, we found two molecules regulated in a species-specific manner, *Shh* and *Medag* up-regulated only in mouse or human PNI respectively (Figs. 2 and 4b, p, Supplementary Dataset 1). Notably, in injured mouse nerves *Shh* levels were high and *Pparg* levels low (Figs. 2 and 4b, c) suggesting reciprocal expression. In adipocytes SHH decreases PPARg activity[62] and a similar mechanism might occur during murine PNI. In human nerves, lack of *SHH* induction would fail to suppress *PPARg* and thereby maintain lipogenic gene expression. Also, MEDAG has been previously attributed a lipogenic function in adipocytes[31]. Interestingly, although the exact function is still unknown, MEDAG positively regulates PPARg and lipogenic genes (e.g. *Fasn*)[31]. Thus, *MEDAG* induction in human but not mouse nerves might help to uphold PPARg levels after injury.

Taken together, our results identify regulation of the lipid metabolism as a novel pathway fundamentally influencing SC reprograming and suggest that this might be a promising target for pharmacological treatment in PNI patients.

## Methods

**Nerve explants.** Murine sural nerves were harvested from C57BL/6 mice. Mice were kept in groups of 2–5 animals in closed cages with food and water ad libitum, 12 h day/12 h night phases, 22 °C temperature and 60% humidity. Mice were killed with $CO_2$ and subsequent cervical dislocation. Thereafter, sural nerves of both hind limbs were dissected by a single cut at each end of the nerve with scissors. One sural nerve serving as control nerve was frozen immediately (0 h time point). The second sural nerve was placed in tubes containing sterile Ringer solution and was incubated at 37 °C for different post-injury time points as indicated. We used 6-month-old male mice supplied by Janvier for histology (Fig. 1, Supplementary Figs. 1 and 12), EM (Fig. 1, Supplementary Fig. 3), gene expression (Fig. 2, Supplementary Fig. 5) and transcriptomic analysis (Figs. 3 and 4). For gene expression comparison between in vivo and ex vivo injury (Supplementary Fig. 2), lipidomic analysis and PIO/DOP/THI/C31 treatment experiments (Figs. 5–7, Supplementary Figs. 10, 11 and 13), 2–3 months old mice of both sexes were used. We observed no overt sex-dependent differences.

Human sural nerves were harvested during reconstruction surgeries in patients with different types of primary nerve injuries (e.g. plexus, *N. medianus* or *N. ulnaris*). Here, the sural nerves were required as auto-transplant in order to bridge these injured nerves and nerve leftovers were used in this study. One piece (~1 cm) of each sural nerve was always frozen on dry ice as soon as possible and served as uninjured control nerve. Depending on the surgical procedure, freezing occurred within 5–30 min (except for EM; see below) after harvesting thereby limiting injury responses as much as possible. Other parts of the sural nerve leftover were immediately cut into 1 cm pieces with a scalpel, placed into tubes with Ringer solution and incubated at 37 °C for different time points. Time between dissection during surgery and freezing was documented for each case.

For murine nerves 4-deoxypyridoxine (DOP; Sigma-Aldrich; dissolved in DMSO; final concentration 1 mM) and pioglitazone (PIO; Sigma-Aldrich; dissolved in DMSO; final concentration 10 µM) treatment was performed by bath application in Ringer solution plus each substance, starting immediately after dissection. For human nerves, PIO (final concentration of 90 µM), GW9662 (GW; Sigma-Aldrich; dissolved in DMSO; final concentration 30 µM) or SR16832 (SR; Tocris; dissolved in DMSO; final concentration 60 µM) was added to injured nerves as soon as possible after harvesting. For 48 h incubation experiments in both, murine and human nerves, the drug containing Ringer solutions were exchanged after 24 h with fresh drug-containing Ringer solution and incubated for another 24 h.

**Sciatic nerve samples ex vivo/in vivo.** Nerve explants of sciatic nerves for the ex vivo injury model were treated the same way as sural nerve explants. The samples of sciatic nerves injured in vivo were kindly provided by the lab or Prof. Simone Di Giovanni (Imperial College London, UK) and derived from mice that had received a sciatic nerve crush.

**Histology.** We fixed nerve explants in 4% FA (formaldehyde) followed by preparation of 5 µm paraffin microtome slices. Immunohistochemistry was performed using Biotin conjugated secondary antibody anti-rabbit (1:500; BA-1000, Vectorlabs) and a peroxidase based detection system using the ABC kit (PK-6100, Vectorlabs) and DAB as substrate. Alternatively, Alexa488 or 546 (1:500; anti-rabbit 488, A-11008; anti-mouse 546, A-11003, Thermo Fisher Scientific) conjugated secondary antibodies were used. Primary antibodies included anti-S100β (rabbit, 1:1000, ab52642, Abcam), anti-βIIITub (mouse, 1:3000, MMS-435P-200, Eurogentec) and anti-cJun (rabbit, 1:500, #9165, Cell signaling).

For teased fibres, murine sural nerves were fixed in 4% FA overnight, teased using forceps and dried on glass slides. Staining was performed using primary antibody anti-Pparg (rabbit, 1:200, ab45036, Abcam) and secondary antibody Alexa488 (1:500; anti-rabbit 488, A11008, Life Technologies).

**Imaging quantification.** Quantification of fluorescent and bright field images was performed using the ImageJ software. For each staining, the function colour threshold was used to set a constant brightness threshold to differentiate between specific staining and background. Depending on the staining either the number (Figs. 1u, v, 6l, and 7e, Supplementary Figs. 1c and 12 b, c) or the area (Figs. 6m, n and 7f, g) of stained objects was quantified using the automated analyse particles function of ImageJ. For all histology, two sections per sample were quantified and the mean value of both sections was used for quantification. For murine samples, one section included a complete cross section of one nerve with an area of ~0.03 µm². For human samples, one nerve section had an area of ~0.09 µm².

For SC quantification (Fig. 1), intact SCs (termed as round SCs) were defined to contain a round ring-like structure with clearly visible borders (for reference see insert in Fig. 1), were associated with a DAPI positive nucleus (not shown in Fig.1) and were counted manually. SCs with distorted morphology and fragments were not included in this quantification.

**Electron microscopy.** Nerve explants were fixed overnight in 4% paraformaldehyde, post-fixed in 2.5% glutaraldehyde for at least 24 h and ultrathin sections were prepared. Four murine nerves (age of mice: 6 months) and four human patient nerves (P28-31; Supplementary Table 1) were used for each time point (0–3 h, 24 h

and 48 h). Human nerves were incubated in fixative as soon as they arrived at our laboratory (here, max. time after dissection was 3 h). For quantification, the myelin sheaths of at least five frames per sample and time point captured with the ×5000 magnification (covering a minimum of 100 myelin sheaths) were counted and the mean was calculated for each sample. Significance was calculated between the means of each sample using an unpaired two-sided *T*-test.

**Quantitative polymerase chain reaction and transcriptomics.** We isolated total RNA from nerves using TRIzol (Qiagen) and the RNeasy kit (Qiagen) according to the manufacturers protocol. Reverse transcription was performed with 0.7 µg RNA using reverse transcriptase (Promega) and random hexamers. We performed qPCR on a Light Cycler 480II (Roche) with the TB Green Premix Ex Taq PCR master mix (Takara). The LC480 II Software detects this threshold cycle value (Ct value) for each sample. Expression of each gene was calculated in relation to RNA levels of the house keeping gene *Hprt* (hypoxanthine phosphoribosyltransferase 1) in order to account for potential variations in total mRNA amounts used for the cDNA synthesis. Primers used are listed in Supplementary Table 2.

For transcriptomics, three murine samples for each time point – 0 h uninjured, 2 h post injury and 24 h post injury – were subjected to microarray analysis. For human samples, five patients (P8, P11, P16, P18 and P19; see Supplementary Table 1) were used for the same time points. RNA was isolated as described above. In all, 100 ng total RNA was used as starting material and 5.5 µg ssDNA per hybridisation (GeneChip Fluidics Station 450; Affymetrix, Santa Clara, CA). The total RNAs were amplified and labelled following the Whole Transcript (WT) Sense Target Labeling Assay (http://www.affymetrix.com). Labelled ssDNA was hybridised to Mouse Gene 2.0 ST or Human Gene 2.0 ST Affymetrix GeneChip arrays (Affymetrix, Santa Clara, CA). The chips were scanned with an Affymetrix GeneChip Scanner 3000 and subsequent images analysed using Affymetrix® Expression Console™ Software (Affymetrix). Raw feature data were normalised and intensity expression summary values for each probe set were calculated using robust multiarray average. Raw feature data were normalised and log2 intensity expression summary values for each probe set were calculated using the robust multiarray average. Differentially expressed genes were determined using BRB-ArrayTools (http://linus.nci.nih.gov/BRB-ArrayTools.html) by a *t*-test and considered statistically significant when $p < 0.05$ and fold change ≥2[63].

**Gene ontology and TF binding motif enrichment analysis.** GO analysis was performed with all genes regulated two-fold or more using the GO Miner Software[64]. TF binding motif enrichment analysis was performed with all genes up- or down-regulated threefold or more using Pscan Ver. 1.5[65].

**Lipidomic analysis.** For lipidomic analysis, all nerves where either frozen immediately, or after 24 h of incubation. The extraction of lipids was carried out using the liquid–liquid extraction protocol[66]. Briefly, internal standards for the selected phospholipids along with sphingosine 1-phopsphate (S1P) d17:1, sphingosine (SPH) d17:1 and ceramide (CER) d18:1/17:0 species and the deuterated PUFAs (AA-d8, EPA-d5, DHA-d5 and DPA-d5; all from Avantis Polaris) were spiked to the tissue samples and lipids were extracted following the protocol[66]. Lipids were analysed by liquid-chromatography multiple reaction monitoring (LC/MRM) using the chromatographic, ionisation and detection conditions[66]. The MRM transitions for the analysis of selected phospholipid, PUFA, sphingolipid and ceramide species were reported[66] with the inclusion of CER (d18:1/16:0) with the MRM transition: *m/z* 538.500 to *m/z* 520.400, and *m/z* 538.500 to *m/z* 264.400 for quantification and qualification, respectively. Calibration curves for the quantification of lipids were acquired using calibration standards and MRM transitions as reported in[66,67] with the inclusion of CER (d18:1/16:0) with the MRM transition: *m/z* 538.500 to *m/z* 520.400 and *m/z* 538.500 to *m/z* 264.400 for quantification and qualification, respectively. Lipids were quantified using Multiquant 3.0.3 software. A heatmap was generated using MetaboAnalyst 4.0[68].

**Statistics and reproducibility.** Numbers (*n*) of independent animal or human samples were indicated in figure bars or text. For statistical analysis of data and graph generation GraphPad Prism software (GraphPad Software, Inc.) was used. Outliers were identified using the ROUT function and included two values in Fig. 2a, c–e, h, three values in Fig. 2f and one value in Fig. 2g (outliers are marked red in the Source Data File). Sample groups were tested for normality using the D'Agostino-Pearson omnibus normality test. Since some groups were not normally distributed, or groups were too small to be tested for normality ($n < 10$), the non-parametric unpaired Mann–Whitney test (two-sided) was chosen to calculate significance if not mentioned otherwise. Statistical significance is provided as *$P \leq$ 0.05, **$P \leq 0.01$ and ***$P \leq 0.001$, respectively. SD is provided if not mentioned otherwise.

For all experiments at least three biological replicates were analysed and exact *n* numbers are indicated in figure legends. Histological analysis in murine tissue (Fig. 1c–h, u, w) was performed with several biological replicates in at least two independent experiments (total $n = 7$ or more). In human tissue (Fig. 1i–n, v, x) every nerve was processed and stained in a separate experiment (total $n = 13$ or more). EM analysis (Fig. 1o–t, y) was performed with four independent biological replicates each for murine and human tissue. qPCR analysis (Figs. 2, 4c–j, p and 7a,

h) for human tissue was performed in a separate experiment for each patient (*n* as indicated in figure legends). For murine nerves (Figs. 2, 4c–j, p and 6a–h), qPCR analysis was performed with several biological replicates as indicated in figure legends (*n* = 4 or more). Histological analysis after PIO treatment was performed with at least four independent biological replicates (as indicated in figure legends) in murine tissue (Fig. 6i–n) and three independent biological replicates in human tissue (Fig. 7b–g).

**Approval of use of human and animal material**. Human samples: all procedures performed using human tissue were approved by the ethics committee of Ulm University. Patients had given their written informed consent to donate not needed nerve tissue prior to surgery. Animal samples: all applicable international, national and/or institutional guidelines for the care and use of animals were followed. This article does not contain any studies with living animals performed by the authors particularly for this project. RNA samples generated from in vivo experiments used in Fig. 4m were kindly provided by the laboratory of Prof. Simone Di Giovanni (Imperial College London, UK). Samples depicted in Supplementary Fig. 1a, d, g were prepared for independent projects in the laboratory of BK and only used as positive controls. Those experiments were approved by the local governmental authority for animal experimentation (Regierungspräsidium Tübingen, Germany).

**Reporting summary**. Further information on research design is available in the Nature Research Reporting Summary linked to this article.

## Data availability

The generated microarray and lipid metabolism datasets are provided as Supplementary Dataset. All source data are provided as a Source Data file. Additional information or data are available upon reasonable request to the Corresponding Author.

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

## Acknowledgements

The work by B.K. has been supported by the Deutsche Forschungsgemeinschaft (DFG) through the Collaborative Research Centre 1149 Danger Response, Disturbance Factors and Regenerative Potential after Acute Trauma (B.K.) and grant KN543/6. Further, B.K. is supported by the Paul und Marlene Hepp-Stiftung and an Ulm University and German Army Hospital research initiative (U2.1d E/U2AD/ED002/EF550). S.M.z.R. acknowledges support by the Baustein programme of Ulm University and Collaborative Research Centre 1149 Danger Response, Disturbance Factors and Regenerative Potential after Acute Trauma. We acknowledge the technical support of Claudia Schwitter for the lipid extraction.

## Author contributions

S.M.z.R. designed, conducted and interpreted experiments and wrote the manuscript. C.B. conducted experiments and performed surgical preparation of human tissue. M.T.P. and G.A. performed surgical preparation of human tissue. J.H. and C.S.S. conducted experiments. R.L. performed part of lipidomic analysis quantification. L.B. designed and performed lipidomic analysis as well as lipidomic data quantification. B.K. designed experiments and co-wrote the manuscript.

## Competing interests

The authors declare no competing interests.
