## [Peer Review File · Nature Communications]

Reviewers' Comments:

Reviewer #1:

Remarks to the Author:

In this manuscript, Reckendorf et al. used an ex vivo injury model of surgery-derived human nerves to directly compare peripheral nerve injury responses of humans and mice. Using their novel experimental system, they observed that human nerve injury responses were delayed or decreased compared to mice, which was likely due to reduced injury-induced adaptation of lipid metabolism in human Schwann cells (SC). This study is an important contribution to the nerve injury field as it has identified lipid metabolism as a key mechanism involved in SC reprogramming in mice, and uncovered molecular differences between mice and humans that may point to novel therapeutic strategies for peripheral nerve injury. Before recommending this manuscript for publication, the following comments should be addressed:

The reviewer appreciates the effort to develop a human model of nerve injury that aims to recapitulate the in vivo situation, and believes it is a worthwhile model for study. However, the authors should provide an explanation, or at least a disclaimer, to address the confound that the human nerves were injured before harvesting. Moreover, the surgical removal of these injured sural nerves from the patients constitutes another nerve injury. Thus, the frozen uninjured control nerve has been subjected to two injuries previously. How does this affect the interpretation of the data?

The authors analyze total RNA from the isolated nerves, but discuss the results as altered gene expression in SCs. They state that '80% of DAPI+ cells are SC' in the nerves, but do not include this data. This data should be shown as it is important to convince readers that the gene expression changes in the nerves are likely occurring in SCs.

One of the main conclusions from this study is that there is a delayed or blunted repair SC induction in injured human nerves compared to murine nerves. As this is the key finding, it would be worthwhile to include analyses at later time-point(s) to determine indeed whether there is a blunted or delayed response.

In the final pharmacological experiments, the authors use the abundance of cJUN as the readout of a repair SC phenotype. As this is an indirect readout, they should at least include analysis of other markers, e.g. Egr2. Moreover, these images should include DAPI co-staining.

In addition to this, the authors are concluding in this manuscript that human SC are blunted in their transition to a repair state. However, c-Jun staining looks comparable in murine and human nerves after 48h without PIO.

Finally, from their transcriptional and lipodomic analyses herein, the authors have found pharmacological methods to suppress the repair SC phenotype in both murine and human nerves. However, as the transition to a repair SC phenotype is the issue in human nerves, can the authors include data, or at least a discussion, on efforts to induce this transition in human SCs.

Reviewer #2:

Remarks to the Author:

The manuscript by Reckendorf et al. presents an interesting novel approach to peripheral nerve injury, which frequently affects humans but is commonly investigated in rodent models. Importantly, the authors investigate species-dependent differences between the repair program of Schwann cells (SC) after injury in mice versus humans (observing an ex vivo paradigm for 48 h after injury). This approach will be of broad interest since the molecular mechanisms leading to the clearly different regeneration potential in rodents versus humans is poorly understood but may well be required for future therapeutic translation.

Generally, all experiments appear well controlled. The authors first validate that their newly established experimental system (ex vivo injury model) preserves cell-to-cell interactions and reproduces early injury responses as seen in vivo. Interestingly, they identify a more pronounced repair SC phenotype in mouse compared to human nerves. By applying genome wide transcriptomics, GO term analysis, transcription factor binding motif enrichment analysis and lipidomics in human vs. mouse SCs they find a strong adaptation of murine SC lipid metabolism after injury which was almost absent in human samples. For murine nerves they discover sphingosine-1-phosphate (S1P) and PPAR γ as candidates to be involved in lipid gene regulation. Using respective inhibitors the authors nicely show that their candidates indeed regulate lipogenic gene expression and that PPAR γ plays a role in SC reprogramming. Interestingly they also find that inhibiting PPAR γ in human SCs led to similar results compared to murine nerves even though a somewhat weaker impact may have been expected owing to the less pronounced downregulation of lipid metabolism in human nerve explants. Further the authors identify (but do not validate) first individual candidates for species specific regeneration differences such as MEDAG, which appear regulated in human but not mouse nerves.

In the reviewer's opinion the manuscript provides not only an attractive new method to assess mouse-to-human differences after nerve injury ex vivo (here mainly focusing on SC biology) but also for straightforward manipulation of the system. The authors further select interesting differences in SC lipid metabolism adaptations after injury between murine and human nerves and validate the role of PPAR γ . They cannot answer why murine SCs initiate lipogenic gene expression and lipidome changes while human SCs do not but speculate about the role of MEDAG. Indeed, MEDAG appears as a plausible candidate but, in this reviewer's opinion, will require some validation in the future. Yet, the manuscript as such appears appropriate for publication in Nature Communications with only minor revisions.

Minor points:

Methods & Statistics:

- Fig 6 and 7: I may have overlooked that, but I did not find information how axonal clearance was quantified (how many, intensity profiles, automated etc.?)
- Fig1: Same goes for SC quantification on IHC, e.g. how do the authors define round SC
- Considering that the authors have gained electron microscopic images, it may be worth quantifying these features on electron microscopic level rather than IHC
- Please improve the definition of "five frames per sample". E.g., how many axons/myelin sheaths where assessed in which condition?
- Have outliers been excluded after testing? If yes, it would be relevant to know whether there were many and where they were excluded
- Unpaired Mann Whitney test: was there a test for non-parametric distribution of data?

Discussion:

- "Downregulation of lipid synthesis genes not been described": Note that Yi et al 2015 (rat nerve crush transcriptome) shows upregulation of lipid-related genes in post-acute phase as well as early LXR/RXR activation (related to Fig3)
- Discussion about synthesis and transport of lipids in nerve regeneration is ongoing for quite some time; the authors may consider referring to that in their discussion

Results section 1/Fig1:

- What is termed myelin outfoldings here probably signifies impaired myelin integrity/shedding. In the neuropathological literature, myelin outfoldings correspond to focal hypermyelination/formation of tomaculae, which is not a prominent feature after nerve injury. I suggest changing the term to impaired integrity
- Line 155-157 (no obvious differences in SC loss after injury): indeed, Fig 1m to this reviewer probably shows more and/or larger SC 48h post injury. It appears important to clarify in methods section how this was quantified. What defines "round" SC? Can quantifying the electron

microscopic images help?

Results section 2/Fig2 and S3:

- Fig2e and S3a: the authors may want to re-check their rel. mRNA abundance data for cJun. In 2e murine (grey) versus human (red) appears roughly similar while in S3a the values for cJun mRNA appear much higher in humans compared to mice
- FigS3: The authors may consider adding boxes with genes in differentiating SCs or repair SCs similar to figure 2
- FigS3e: BRN2 capital lettering means human data?

Fig 3:

- i: which TF motifs come up at 2h?

Fig S4:

- legend: transcriptomics typo

Results section Fig 4:

- line 359 "weak downregulation in human nerves (4c-j)" for some of genes (Pparg, Sreb, Fasn): I do not see a downregulation in human samples. Significance?
- The authors may wish to re-check their data for Pparg in human samples; here downregulation was detected by transcriptomics 24h after injury but by qRT PCR "recovered" by 48h
- Teased fiber preparations: For reference, it would be relevant to see additional markers in the compilation, in particular classical axonal and myelin markers
- Line 371: "in keeping with": this sentence doesn't make much sense; please rephrase
- I may have overlooked the corresponding statement; what is the age of the 5 patients of which samples were analyzed here

Fig5:

- While 0h and 24 h cluster nicely next to each other for murine samples, timepoints do not cluster well for the human samples. Is that a consequence of the calculation?

Results section 6/Fig6:

- line 502-506: PIO induced Mbp expression which shows it favors myelinating (not repair) SC; however the next sentence is influence of PIO on repair SC phenotype "induction". Please clarify. Is "induction" the best possible wording?
- Lines 501-514 are not well phrased compared to the other text; the authors may wish to consider improving the phrasing

Results section 7/Fig7:

- SREBP and ACSL1 do not react on PIO treatment in human but in murine nerves. Any idea why?
- Axon clearance diminished in human nerves treated with PIO (Fig7 c and e): The statement is somewhat daring and may easily be overinterpreted by some readers as long as solely based on IHC. Electron microscopy may provide a much more solid data basis

Supplemental table S1:

- I may have overlooked the information - which patient samples were used in transcriptomics, in particular it appears relevant to know the age distribution

MEDAG: The authors may consider whether it is possible to determine MEDAG expression in older vs younger patients considering that it came up as "human" specific candidate but is involved in adipogenesis. Is it possible that its expression in humans is related to age? At least, the authors may consider testing its expression in "fit/young" compared to "fat/old" mice

Reviewer #3:

Remarks to the Author:

In the present manuscript by Mayer zu Reckendorf and colleagues investigate the regulation of mechanisms that are activated in response to nerve injury and identify potential targets that are different between mice and humans using an elegant ex vivo model. They demonstrate that both lipid levels as well as gene expression profiles are very different in murine when compared to human Schwann cells. They then identify the S1P pathway as well as PPAR γ as potential targets in regulating Schwann cell de-differentiation and accelerating repair processes and attempt to modulate these pathways using pharmacological approaches. The experiments appear to be well conducted and the topic of investigation together with the approaches employed are very novel. The manuscript is however largely descriptive and with the approaches used to demonstrate cause and effect being somewhat weak, thus limiting the enthusiasm towards the study. In this regard there are the following specific comments that the authors would need to address:

- 1) In the title as well as throughout the manuscript the authors refer to lipid metabolism, however the data presented at presents abundance of a small group of lipid species therefore the authors should 1) explain why they focused on these lipid species, what is their relevance to the biology of Schwann cells 2) revise the text to reflect the results presented or 3) indeed conduct experiments to demonstrate the flux of lipids in these cells
- 2) The methods section referring to the metabolomics is incomplete given that the reference they provide points to another 3 different references which in themselves do not provide detailed methods. The authors should provide sufficient details of the methods that will allow evaluation of the methods
- 3) While the authors provide information on age for the mice there is no reference to whether males and females were used for the experiments presented. This is an important aspect since while in humans there do not appear to be big sex-related differences this may not be the case in mice.
- 4) In the transcriptomic data presented in Figure 3 given that part of the argument that the authors put forward is that the inflammatory mechanisms are unaltered between mice and humans the authors need to validate at least a subset of the genes given that transcriptomics may yield false positive/negative results.
- 5) The evidence for the role of S1P in the proposed mechanism is inconclusive given that as the authors themselves state the inhibitor used is a non-specific inhibitor. Therefore, since the authors put a significant emphasis on the role of S1P in the proposed mechanism they need to provide additional evidence using S1PR agonists/antagonists and/or addback of S1P.
- 6) Another aspect that is not well developed is the role of PPAR γ in regulating human Schwann cell responses. The evidence presented for a role of PPAR γ elicited signalling in mice is compelling however whether this is indeed the 'defect' in human cells is weak. The authors need to demonstrate that via the activation of PPAR γ there is indeed an improvement of nerve repair.

Reviewer #1

(changes to the manuscript are highlighted in red)

1) *The reviewer appreciates the effort to develop a human model of nerve injury that aims to recapitulate the in vivo situation, and believes it is a worthwhile model for study. However, the authors should provide an explanation, or at least a disclaimer, to address the confound that the human nerves were injured before harvesting. Moreover, the surgical removal of these injured sural nerves from the patients constitutes another nerve injury. Thus, the frozen uninjured control nerve has been subjected to two injuries previously. How does this affect the interpretation of the data?*

There might have been a misunderstanding in the manuscript wording for this part.

The patients that were included in this study had different types of peripheral nerve injury (e.g. plexus, N. medianus or N. ulnaris injuries). However, in all patients the sural nerve was never the primary injury site and was only harvested to bridge the nerve gap for the primary injured nerve. Thus, the first injury that was applied to the sural nerves in our study was for harvesting this donor nerve during surgery. After harvesting the sural nerve, a piece of the leftover of it not required as autograft was frozen straight away and served as control nerve. Thus, the control nerves were injured only once and directly frozen. Depending on the surgical procedure, the nerve pieces serving as “control nerves” in our experiments were typically frozen within 5-30 minutes so that no major injury responses could occur.

The remaining sural nerve piece was then cut into smaller pieces (approximately 1 cm long) and incubated for different time-points (2 h, 24 h and 48 h) to monitor injury responses. Thus, only these nerves reflecting the injury condition were cut twice.

We have rephrased this part of the methods section of the manuscript in order to make it more clear for the reader (p. 35).

2) *The authors analyze total RNA from the isolated nerves, but discuss the results as altered gene expression in SCs. They state that ‘80% of DAPI+ cells are SC’ in the nerves, but do not include this data. This data should be shown as it is important to convince readers that the gene expression changes in the nerves are likely occurring in SCs.*

We now have added a quantification of this data to the revised Fig. S1 and have indicated this also in the results section of the main text (p. 5).

3) *One of the main conclusions from this study is that there is a delayed or blunted repair SC induction in injured human nerves compared to murine nerves. As this is the key finding, it would be worthwhile to include analyses at later time-point(s) to determine indeed whether there is a blunted or delayed response.*

We agree with the reviewer that this was an interesting open question in our manuscript. In order to investigate this point, we incubated human nerves for up to five days upon injury and analysed lipogenic gene expression. This analysis revealed that expression of genes coding for proteins involved in fatty acid anabolism (SREBP, ACACA, ACSL1, FASN, DGAT2) eventually also reached similar low levels as obtained in mice. However, in mice these low levels were reached already at two days post injury. Interestingly, genes related to fatty acid catabolism (ECHS1, EHHADH) became induced 3-5 days after injury.

Together, those results indicate that adaptation of lipid metabolism in human nerves is rather delayed than blunted in human nerves. The results are depicted in the revised Fig. S8 and commented in the Results section of the manuscript (p. 15-16).

4) *In the final pharmacological experiments, the authors use the abundance of cJUN as the readout of a repair SC phenotype. As this is an indirect readout, they should at least include analysis of other markers, e.g. Egr2. Moreover, these images should include DAPI co-staining. In addition to this, the authors are concluding in this manuscript that human SC are blunted in their transition to a repair state. However, c-Jun staining looks comparable in murine and human nerves after 48h without PIO.*

cJUN is the prototypical TF activated in repair SCs whereas EGR2 is present in myelinating SCs. We tried to stain EGR2 to visualise myelinating SCs as proposed by the reviewer, but

none of the two rabbit polyclonal antibodies (NovusBio, Biozol) worked on paraffin sections. In order to address this comment, we now have included MBP staining and quantification (Fig. 6 and 7). MBP is expressed in myelinating SCs and will be downregulated on mRNA level and degraded on protein level in repair SCs. Our data show that MBP is indeed downregulated on mRNA and protein level in injured nerves and PIO treatment hinders this downregulation (Fig. 6 and 7).

Further, the reviewer suggests to use DAPI co-staining in order to show nuclear localization of cJUN in injured nerves. We have now included fluorescent cJUN/DAPI staining in Fig. S12. Additionally, we have included quantification of DAPI⁺ nuclei in uninjured murine and human nerves and nerves 48 h post injury with control or PIO treatment. This quantification proves, that cJUN downregulation in PIO treated nerves is not due to a decline in nuclei numbers.

Regarding cJUN expression upon injury, we agree with the reviewer that the levels are comparable in human and murine nerves (Fig. 6 and 7). Of note, this is also the case on mRNA level (Fig. 2e). cJUN upregulation after injury is a prerequisite for the transition of myelinating to repair SCs. Hence, the fact that cJUN gets upregulated in murine and human nerves shows that human SCs in principle have the capability to become repair SCs. We hypothesise that the delay in human SC reprogramming probably relies on some downstream process, not directly involving cJUN upregulation.

5) Finally, from their transcriptional and lipodomic analyses herein, the authors have found pharmacological methods to suppress the repair SC phenotype in both murine and human nerves. However, as the transition to a repair SC phenotype is the issue in human nerves, can the authors include data, or at least a discussion, on efforts to induce this transition in human SCs.

The reviewer is right that in the initial phase after nerve injury in humans, it would be beneficial to identify pharmacological approaches to induce SC reprogramming. Since we hypothesize that PPAR γ activity and as a result lipogenic gene expression are key mechanisms in SC reprogramming, we now treated human nerves with two different PPAR γ antagonists (GW9662 and SR16832) in order to block its activity. Indeed, repression of PPAR γ activity resulted in decreased lipogenic gene expression. We have included those data in the revised Fig. 7f with the corresponding text in the Results (p. 28) and Discussion section (p. 33). Because of the limited amount of human material available during the revision period, unfortunately we were not able to perform additional histological analysis to evaluate actual SC reprogramming. Nevertheless, considering the data obtained from mouse and human nerves, there is a strong line of evidence suggesting that downregulation of lipogenic genes will actually result in accelerated SC reprogramming.

Reviewer #2:

(changes to the manuscript are highlighted in red)

Minor points:

Methods & Statistics:

1.) *Fig 6 and 7: I may have overlooked that, but I did not find information how axonal clearance was quantified (how many, intensity profiles, automated etc.?)*

Fig1: Same goes for SC quantification on IHC, e.g. how do the authors define round SC

We have added the information about imaging quantification in a separate chapter in the Materials and Methods part (p. 37)

2.) *Considering that the authors have gained electron microscopic images, it may be worth quantifying these features on electron microscopic level rather than IHC*

We tried to address Schwann cell morphology the electron microscopy pictures. Unfortunately, ultrathin sections prepared for electron microscopy only contain parts or fragments of each Schwann cell, which makes it impossible to effectively quantify the entire Schwann cell morphology.

Nevertheless, we now used EM pictures in addition to histology in order assess axonal clearance. We have added these data in new Fig. S3.

3.) *Please improve the definition of “five frames per sample”. E.g., how many axons/myelin sheaths were assessed in which condition?*

We have added this information in the Electron Microscopy section of the Materials and Methods (p. 38).

4.) *Have outliers been excluded after testing? If yes, it would be relevant to know whether there were many and where they were excluded*

We have added this information in the Statistics section of the Materials and Methods.

5.) *Unpaired Mann Whitney test: was there a test for non-parametric distribution of data?*

We performed a normality test in order to evaluate data distribution. In some cases, the data did not pass the normality test. In other cases, groups are too small to perform a normality test ($n < 10$). For this reason, we decided to use the Mann-Whitney test over the parametric T-test. We have now added this information in the Statistics section of the Materials and Methods (p. 40-41). Since the Mann-Whitney test is much more stringent than the T-test, we believe that this even strengthens the power of our results.

Discussion:

6.) *“Downregulation of lipid synthesis genes not been described”: Note that Yi et al 2015 (rat nerve crush transcriptome) shows upregulation of lipid-related genes in post-acute phase as well as early LXR/RXR activation (related to Fig3)*

We have now included this point in our Discussion (p. 31).

7.) *Discussion about synthesis and transport of lipids in nerve regeneration is ongoing for quite some time; the authors may consider referring to that in their discussion*

We have now included this point in our Discussion (p. 31).

Results section 1/Fig1:

8.) *What is termed myelin outfoldings here probably signifies impaired myelin integrity/shedding. In the neuropathological literature, myelin outfoldings correspond to focal hypermyelination/formation of tomaculae, which is not a prominent feature after nerve injury. I suggest changing the term to impaired integrity*

The reviewer is right about the term “outfoldings”. We have replaced it by “impaired integrity” as suggested (p. 6).

9.) Line 155-157 (no obvious differences in SC loss after injury): indeed, Fig 1m to this reviewer probably shows more and/or larger SC 48h post injury. It appears important to clarify in methods section how this was quantified. What defines “round” SC? Can quantifying the electron microscopic images help?

Regarding the area, SCs indeed seem to become larger upon injury. The reason for this is most likely the myelin integrity loss which makes SCs appear “swollen”. We included additional text (p. 6) and edited the inserts in Fig. 1d, g, j, m in order to explain how SCs were quantified. In addition, we describe the quantification procedure in the new Imaging Quantification chapter in the Methods (p. 37-38). As described above (see point 2), electron microscopy sections were not well-suited for the quantification of SC morphology.

Results section 2/ Fig2 and S3:

10.) Fig2e and S3a: the authors may want to re-check their rel. mRNA abundance data for *cJun*. In 2e murine (grey) versus human (red) appears roughly similar while in S3a the values for *cJun* mRNA appear much higher in humans compared to mice

This difference is due to different normalization of data in both figures.

In Fig. 2, the relative mRNA expression of injured nerves was normalized to expression levels for uninjured nerves. Here, mRNA levels in uninjured nerves were set to 1 for each gene and the fold change was calculated for the other post-injury time points. This normalization allowed for directly comparing fold changes between mouse and human nerves. Since we had to use different primer pairs for human and murine samples, interpretation on absolute mRNA abundance has to be taken with a note of caution.

In Fig. S3, values were not normalized to the uninjured nerve since we do not directly compare mouse and human samples but only either murine or human samples with each other.

11.) FigS3: The authors may consider adding boxes with genes in differentiating SCs or repair SCs similar to figure 2

We appreciate this comment and have added the boxes as suggested which helps to understand the figure more readily.

12.) FigS3e: *BRN2* capital lettering means human data?

The lettering for *BRN2* was corrected as suggested (new Fig. S3).

13.) Fig 3: i: which TF motifs come up at 2h?

We used differentially expressed genes to perform the TF binding motif analysis for each time point for human and murine samples. Since, at 2 h there were no differentially regulated genes in human samples (see Fig. 2a) we could not perform this analysis.

14.) Fig S4: legend: transcriptomics typo

We have corrected the typo.

Results section 4/ Fig4:

15.) line 359 “weak downregulation in human nerves (4c-j)” for some of genes (*Pparg*, *Sreb*, *Fasn*): I do not see a downregulation in human samples. Significance?

There is indeed a weak but significant downregulation of lipogenic genes also in human samples at 24 h. We have added the significance for each chart in the figure using coloured asterisks.

16.) The authors may wish to re-check their data for *Pparg* in human samples; here downregulation was detected by transcriptomics 24h after injury but by qRT PCR “recovered” by 48h

At 24h after injury, data on a slight *PPARg* downregulation in human nerves are congruent between qPCR data (Fig. 4c) and transcriptomics (Fig. 4b). However, the reviewer is right that at 48 h after injury (only tested in qPCR) *PPARg* mRNA was elevated again at 48 h. Interestingly, several other lipogenic genes validated by qPCR (Fig. 4c-j) followed the same pattern. In response to Reviewer 1 we analysed human nerves up to 5 days post injury with

qPCR (new Fig. S8). Overall we observed that several lipogenic genes showed a downregulation after injury however with a clear temporal delay compared to mouse nerves. In contrast to those genes, PPAR γ did not show a consistent downregulation and expression levels were either slightly downregulated or comparable to the pre-injury mRNA levels.

17.) Teased fiber preparations: For reference, it would be relevant to see additional markers in the compilation, in particular classical axonal and myelin markers

We now added the SC specific S100 β staining in the merged picture (Fig. 4k) in order to demonstrate that the observed PPAR γ staining is indeed in SC nuclei.

18.) Line 371: "in keeping with": this sentence doesn't make much sense; please rephrase
We rephrased this sentence for better understanding.

19.) I may have overlooked the corresponding statement; what is the age of the 5 patients of which samples were analyzed here

We agree that age is an important parameter. The age of each patient and the experiments the patient samples were used for are listed in Table S1.

20.) Fig5: While 0h and 24 h cluster nicely next to each other for murine samples, timepoints do not cluster well for the human samples. Is that a consequence of the calculation?

The non-clustering of human samples is indeed a consequence of calculation. We used lipid levels for all indicated lipids in Fig. 5a before and 24 h after injury for five human and five murine samples. Those data were analysed using the MetaboAnalyst 4.0 software as indicated in the Methods section. The software analyses to what extent lipid profiles are similar between different samples and orders the samples according to similarity.

The fact that human samples at 0 h or 24 h do not cluster together is actually perfectly in line with our hypothesis that the lipid profile upon injury does not significantly change in human nerves. In contrast, murine samples show consistent and significant changes resulting in tight clustering of the 0 h samples together and clear segregation from the 24 h samples which cluster separately.

Results section 6/Fig6:

21.) line 502-506: PIO induced Mbp expression which shows it favors myelinating (not repair) SC; however the next sentence is influence of PIO on repair SC phenotype "induction". Please clarify. Is "induction" the best possible wording?

The reviewer is right that the term "induction" is misleading at this point. We have now rephrased the sentence.

22.) Lines 501-514 are not well phrased compared to the other text; the authors may wish to consider improving the phrasing

We have rephrased this part in order to make it more comprehensive to the reader (p. 24).

Results section 7/ Fig. 7:

23.) SREBP and ACSL1 do not react on PIO treatment in human but in murine nerves. Any idea why?

Indeed, SREBP and ACSL1 expression showed no significant reaction to PIO treatment. Nevertheless, SREBP mean expression was actually twice as high upon PIO treatment compared to control treated samples. Yet no significance was reached due to the variability between human samples and the relatively stringent Mann-Whitney test. When run through a T-test, we get $p = 0.0487$. ACSL1 expression does certainly not react to PIO treatment in human samples. Interestingly, this is also the only gene in murine samples that was least affected by PIO treatment (see Fig. 6b). Currently, we do not have an explanation why this gene reacts differently than other lipogenic genes. One possibility is that other transcription factors besides PPAR γ that appeared in our study such as RXRA (Fig. 4n) or SREBP1 (Fig. 4b) are involved in transcriptional regulation of those genes.

24.) *Axon clearance diminished in human nerves treated with PIO (Fig7 c and e): The statement is somewhat daring and may easily be overinterpreted by some readers as long as solely based on IHC. Electron microscopy may provide a much more solid data basis*

Histological quantification of axon clearance was used in Fig. 6 (murine nerves) and Fig. 7 (human nerves). We now additionally provide electron microscopy analysis of PIO treated injured murine nerves for the evaluation of axon degradation/clearance (new Fig. S13). Due to the limited human material available during the revision period, we could not provide those data for human nerves. Nevertheless, in murine nerves EM analysis is in line with histological analysis, so we expect that this would also be the case in human nerves.

25.) *Supplemental table S1: I may have overlooked the information - which patient samples were used in transcriptomics, in particular it appears relevant to know the age distribution*

The age of the patients used for different kind of analyses is listed in Table S1. Specifically, patients used in transcriptomics are listed in the column "MA" (microarray).

26.) *MEDAG: The authors may consider whether it is possible to determine MEDAG expression in older vs younger patients considering that it came up as "human" specific candidate but is involved in adipogenesis. Is it possible that its expression in humans is related to age? At least, the authors may consider testing its expression in "fit/young" compared to "fat/old" mice*

We tested this suggestion but did not observe an age-dependent expression of MEDAG between nerves of younger and older patients.

Reviewer #3:

(changes to the manuscript are highlighted in red)

1.) *In the title as well as throughout the manuscript the authors refer to lipid metabolism, however the data presented at presents abundance of a small group of lipid species therefore the authors should 1) explain why they focused on these lipid species, what is their relevance to the biology of Schwann cells 2) revise the text to reflect the results presented or 3) indeed conduct experiments to demonstrate the flux of lipids in these cells*

In the first part of the manuscript we performed an unbiased genome-wide search for genes modified in injured nerves. Here, we observed downregulation of genes encoding for enzymes of the lipid metabolism modulating abundance of many lipid species including fatty acids, triacyl glycerides, eicosanoids and e.g. sphingolipids. In the subsequent lipidomic part we focused so far mainly on glycerophospholipids and sphingolipids (Fig. 4). Particularly sphingolipids were interesting for this study since this lipid species has signalling potential and was suggested to affect myelinating cell function. Indeed, our lipidomic analysis revealed alterations for several such sphingolipids.

In order to expand on the lipidomics analysis we now performed more experiments and included several PUFAs such as EPA, DHA, AA and DPA. In general, we did not observe major changes in the abundance of these selected PUFAs after injury in human or mouse nerves (new Fig. S9). This is perhaps somewhat surprising given that many genes encoding enzymes for fatty acid synthesis or degradation were affected on transcript level (Fig. 4). However, it should be kept in mind that our lipidomic analysis on PUFAs only covered a very small subset of all PUFAs existing. We could not include more fatty acid species (e.g. saturated and mono-unsaturated fatty acids) since mass spectrometry based detection was not established for more lipids.

We now revised the text to more clearly explain the reasoning of choosing particularly sphingolipids for lipidomics and their relevance to SCs. Furthermore, we included more wording for the new data on PUFAs.

2.) *The methods section referring to the metabolomics is incomplete given that the reference they provide points to another 3 different references which in themselves do not provide detailed methods. The authors should provide sufficient details of the methods that will allow evaluation of the methods*

We have revised the lipidomic part of the Material and Methods in order to provide full information about how the analysis was performed.

3.) *While the authors provide information on age for the mice there is no reference to whether males and females were used for the experiments presented. This is an important aspect since while in humans there do not appear to be big sex-related differences this may not be the case in mice.*

In all experiments where “young” (2-3 months old) mice were used (Figures 5, S2, S10-S12), both sexes were employed for analysis and we could not detect any sex-dependent differences. This was similar to human nerves, where we could not detect sex-dependent differences either. Experiments with “old” (6 months old) mice were conducted only with male mice (Figures 1-4, 6, S1, S3, S5) since Janvier delivers only male mice at this age.

We have rephrased the relevant text in the Materials and Methods section in order to include this information (p. 35).

4.) *In the transcriptomic data presented in Figure 3 given that part of the argument that the authors put forward is that the inflammatory mechanisms are unaltered between mice and humans the authors need to validate at least a subset of the genes given that transcriptomics may be yield false positive/negative results.*

We now tested several inflammatory genes (*Rel*, *Ccl2*, *Ccl7*, *Cxcl10*, *Il1 β* , *Ptgs2*) altered in our transcriptomics data (Fig. 3) in independent qPCR experiments. We observed an almost identical induction of all inflammatory genes after injury between mouse and human nerves

thus corroborating our transcriptomics data. This data set is now included in the revised version of the manuscript with more wording (p. 12) and the new Fig S5.

5.) *The evidence for the role of S1P in the proposed mechanism is inconclusive given that as the authors themselves state the inhibitor used is a non-specific inhibitor. Therefore, since the authors put a significant emphasis on the role of S1P in the proposed mechanism they need to provide additional evidence using S1PR agonists/antagonists and/or addback of S1P.*

DOP is indeed a non-specific inhibitor for the S1P lyase, yet it is a widely used inhibitor for this purpose. Nevertheless, we agree that off-target effects could occur while using it. In order to address this concern, we repeated the relevant experiments now using two other specific inhibitors, THI (2-acetyl-5-tetrahydrobutyl imidazole) and compound 31. Both inhibitors were reported in the literature before to inhibit S1P lyase and thereby modulate S1P signalling (respective references are now included in the manuscript). For both inhibitors, we could observe that S1P lyase inhibition after injury resulted in significantly elevated lipogenic gene expression compared to control treated injured nerves. For Pparg, we observed a full rescue of mRNA abundance whereas for most other genes, both inhibitors approximately doubled mRNA levels compared to injured nerves without inhibitor treatment (new Fig. S11). Overall, it is fair to mention that compared to DOP, effects of both inhibitors were weaker. Nevertheless, we now provide three independent means of S1P lyase inhibition all pointing towards the same direction.

We applied commercially available S1P to the nerve culture medium as suggested by this Reviewer. However, no reproducible effects were observed in these experiments.

6.) *Another aspect that is not well developed is the role of PPARg in regulating human Schwann cell responses. The evidence presented for a role of PPARg elicited signalling in mice is compelling however whether this is indeed the 'defect' in human cells is weak. The authors need to demonstrate that via the activation of PPARg there is indeed an improvement of nerve repair.*

We are not entirely sure, that we have interpreted this comment of the reviewer correctly.

We had already performed experiments employing “activation of PPARg” in the original version of the manuscript. Here, PPARg activation by pioglitazone (PIO) enhanced mRNA abundance of lipid metabolism encoding genes after injury in mouse (Fig. 6). Importantly, with regard to the reviewers point these experiments had already also been performed in human nerves (Fig. 7).

We now more thoroughly inspected PPARg expression in human nerves after injury. For this, we analysed more post-injury time points in human nerves for PPARg mRNA expression (new Fig. S8). Indeed, also up to five days after injury, PPARg levels in human nerves never dropped as strongly as observed in mouse nerves already at 2 days post injury (new Fig. S8). Taken together, since PPARg expression in human SCs only mildly decreases after injury, we believe that this might at least in part account for a delay in human SC reprogramming compared to rodent nerves.

Nevertheless, in order to strengthen our data on the role of PPARg we now also included PPARg antagonists (GW9662 and SR16832) on injured human nerves in order to suppress PPARg activity. Indeed, PPARg inhibition resulted in decreased lipogenic gene expression after injury (new Fig. 7h). This new data set complements our findings on PIO mediated PPARg activation where we observed induction of these genes.

Unfortunately, our human ex vivo system does not allow us to investigate neuronal responses and thus we cannot directly investigate an effect on “nerve repair”. This is due to the fact that dissected nerves do not include neuronal cell bodies precluding analysis of de novo axonal outgrowth. Nevertheless, our human nerve culture system is well-suited to monitor SC responses and investigate the impact of pharmacological manipulations on SC responses as a first proof-of-principle experiment towards possible clinical trials in human PNI patients. Indeed, our study identified several new pharmacological compounds (e.g. DOP, PIO, etc.) involved in SC reprogramming which are FDA-approved for other treatments which might serve as starting point towards more translational approaches.

Reviewers' Comments:

Reviewer #1:

Remarks to the Author:

In this revised manuscript, the authors have addressed concerns I raised during previous review.

Reviewer #2:

Remarks to the Author:

All comments have been adequately considered by the authors.

This is a nice paper, congratulations.

Reviewer #3:

Remarks to the Author:

The authors have addressed my comments in full. The manuscript has significantly improved.